# G²TAM: Geometry Grounded Track Anything Model

Chenming Zhu [1 2]  Peizhou Cao [2 3]  Jingli Lin [2 4]  Wenbo Hu [2 5]  Yunlong Ran [2 6]  Jiangmiao Pang [2]
Tai Wang [2]  Xihui Liu [1]

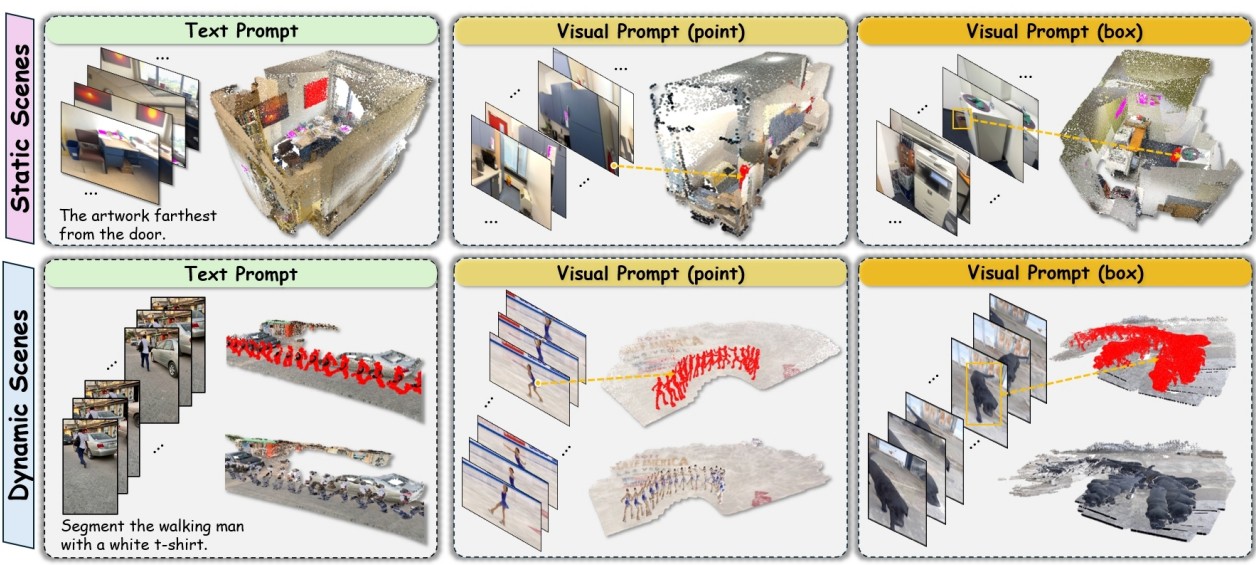

*Figure 1.* Given unordered images or video inputs, **G²TAM** supports various prompts—including text, point, and box prompts—to perform joint 3D reconstruction and spatial–temporal consistent instance segmentation, enabling promptable instance tracking in 3D space.

## Abstract

Human spatial understanding arises from jointly perceiving geometry and semantics, enabling consistent object identification and localization across viewpoints and time. Current video segmentation models depend on explicit object appearance memory banks for instance tracking, yet they remain vulnerable to large viewpoint changes and long-term occlusions. Leveraging the spatial consistency afforded by modern feed-forward 3D reconstruction models, we propose the **Geometry Grounded Tracking Anything Model (G²TAM)**, a unified framework for promptable

instance tracking in 3D using only unordered RGB images or videos. G²TAM employs spatially aligned geometric representations as implicit memory, ensuring stable instance identity and localization across frames and views. At its core is a cross-modal spatial encoder that integrates visual and textual prompts into a shared geometric space, enabling end-to-end spatial reconstruction and instance-consistent mask prediction. To support training and evaluation, we construct **InsTrack**, a large-scale dataset with a dedicated validation split for benchmarking. Extensive experiments show that G²TAM delivers strong cross-view consistency, promptable instance spatial tracking, video object segmentation and spatial reconstruction, establishing a foundation for interactive, geometry-grounded spatial reasoning.

[1]The University of Hong Kong [2]Shanghai AI Laboratory [3]Beihang University [4]Shanghai Jiao Tong University [5]University of California, Los Angeles [6]Zhejiang University. Correspondence to: Xihui Liu <xihuiliu@hku.hk>.

*Proceedings of the 43rd International Conference on Machine Learning*, Seoul, South Korea. PMLR 306, 2026. Copyright 2026 by the author(s).

# 1. Introduction

Humans recognize and track objects not by relying on appearance alone, but by integrating visual cues with a stable sense of 3D structure. Even when an object's appearance changes due to lighting, deformation, or drastic viewpoint shifts, humans maintain identity by grounding perception in a coherent spatial understanding of the scene. This complementary use of appearance and geometry enables reliable tracking in both static environments and dynamic, motion-heavy scenarios.

Current video object segmentation (VOS) methods (Ravi et al., 2024; Kirillov et al., 2023; Li et al., 2023) typically rely on appearance matching and explicit memory banks to perform temporal-consistent video object tracking, which often falter under drastic viewpoint shifts or long-term occlusions. While recent efforts (Plizzari et al., 2025; Bhalgat et al., 2024) attempt to incorporate 3D geometry information into egocentric tracking to achieve object permanence even when objects are out-of-view. These methods typically follow a "Tracking-by-Mapping" paradigm, which performs post-hoc matching by projecting explicit 2D tracking or detection results into 3D space via off-the-shelf pose and depth estimators. Such multi-stage pipelines may inherently suffer from error propagation between disconnected 2D and 3D modules and are strictly limited to scenarios where accurate geometry information is available.

With the recent progress in feed-forward 3D reconstruction (Wang et al., 2024; 2025a), which allows for inferring aligned geometry directly from unposed images, a fundamental question arises: *Should geometry remain a mere post-hoc "checker" for verification, or can it serve as a "latent foundation" that implicitly grounds the entire perception process?*

We introduce the **Geometry-Grounded Tracking Anything Model (G²TAM)**, a unified end-to-end framework for promptable instance tracking in 3D space, taking only unordered images or video as input. At the core of G²TAM is the concept of *Geometry as Implicit Memory*: Unlike prior works that rely on explicit temporal memory bank or further post-hoc verification using explicit geometric representations (e.g., depth maps and camera poses), we are the first to unify appearance and geometry within a latent, spatially-aligned feature space to serve as the underlying persistent memory for identity reasoning. To enable seamless support for multi-modal prompting, G²TAM embeds text/visual prompts directly into a unified geometric semantic representation through a **simple yet highly effective cross-modal spatial encoder**. This early-fusion design ensures that recognition and identity reasoning are inherently grounded in the same spatial structure—leading to more precise local recognition and robust cross-view identity stability. Paired with lightweight geometry and mask decoders,

G²TAM simultaneously performs spatial reconstruction and spatially-consistent instance tracking across large viewpoint changes and complex dynamic scenes—achieving robust 3D-aware tracking without any explicit 3D geometry inputs and temporal memory banks.

Our experimental results demonstrate that the joint training of reconstruction and segmentation significantly enhances cross-view mask consistency, effectively validating geometry as a persistent form of implicit memory. The resulting geometry-aware representation exhibits superior generalization across diverse tasks—including promptable instance spatial tracking, 3D visual grounding, and video object segmentation—maintaining robust performance even under extreme camera motion and dynamic complexity. Notably, G²TAM achieves 74.3 S-mIoU on the instance spatial tracking benchmark, outperforming the state-of-the-art SAM2 (Ravi et al., 2024) (47.6 S-mIoU) by a substantial margin and setting a new milestone for spatially-consistent tracking.

# 2. Related Work

## 2.1. Geometry Foundation Models

Traditional 3D reconstruction (Schönberger & Frahm, 2016; Hartley & Zisserman, 2003; Pan et al., 2024; Schönberger & Frahm, 2016; Furukawa et al., 2015; Schönberger et al., 2016) relies on multi-view geometry for feature matching and pose estimation. Recently, feed-forward models have emerged to directly regress the 3D structure of a scene from a set of images in a single pass. Dust3R (Wang et al., 2024) predicts point clouds from pairs of images within the reference camera's coordinate system. VGGT (Wang et al., 2025a) addresses this issue by incorporating multi-task learning and training on large-scale datasets. Pi3 (Wang et al., 2025c) further employs a fully permutation-equivariant architecture to predict affine-invariant camera poses and scale-invariant local point maps without any reference frames. However, these models remain geometry-centric, excelling at spatial reconstruction but lacking the semantic adaptability and interactive capability required in real-world scenarios.

## 2.2. Promptable Video Object Segmentation

Current VOS methods follow two main paradigms: (1) Semi-supervised VOS (Semi-VOS), which tracks objects based on an initial mask (Cheng & Schwing, 2022; Bekuzarov et al., 2023; Yang et al., 2021; Yang & Yang, 2022), with SAM2 (Ravi et al., 2024) delivering strong generalization and enabling interactive segmentation through a unified prompting framework. (2) Referring VOS (RVOS), which segments objects via language descriptions (Botach et al., 2022; Wu et al., 2022; Liang et al., 2025). Despite these advances,

current VOS models remain appearance-driven and rely on temporal matching or feature banks, making them sensitive to large camera motions and viewpoint changes. In contrast, our G²TAM leverages geometry as implicit memory, enabling consistent object segmentation across challenging views and long-term spatial variations.

## 2.3. 3D-Aware Instance Tracking

Recent efforts in egocentric vision have focused on achieving object permanence during drastic camera motion and out-of-view occlusions. LMK (Plizzari et al., 2025) introduces the OSNOM task and proposed a "Lift, Match, and Keep" framework that projects 2D observations into 3D world coordinates for post-hoc matching. Building upon this, the subsequent work (Bhalgat et al., 2024) integrates scene geometry with 2D video object segmentation (VOS) models to refine tracking consistency, which leverages 3D awareness to re-identify objects that have been out of sight for extended periods. While effective, these methods follow a multi-stage "Tracking-by-Mapping" paradigm, relying on disconnected off-the-shelf pose and depth estimators which are prone to error propagation. In contrast, G²TAM avoids such explicit pipelines by unifying appearance and geometry within a latent, spatially-aligned feature space, enabling end-to-end identity reasoning without explicit 3D inputs.

## 3. Method

In this section, we first outline the architecture of G²TAM (Sec. 3.1 and 3.2), followed by the construction pipeline and statistics of the *InsTrack* dataset and benchmark (Sec. 3.3). Finally, we detail the model's training objectives (Sec. 3.4).

### 3.1. Problem Formulation

The input to our model is a sequence $\mathcal{I} = (\mathbf{I}_i)_{i=1}^N$ of $N$ RGB images $\mathbf{I}_i \in \mathbb{R}^{H \times W \times 3}$ from static or dynamic scenes. The model is conditioned on a prompt $P \in \{P_v, P_t\}$, where $P_v$ represents visual prompts (points or boxes) and $P_t$ represents text prompts. Our G²TAM is a function $\Phi$ that maps the image sequence and the prompt to a corresponding set of 3D geometry and consistent instance masks:

$$\Phi((\mathbf{I}_i)_{i=1}^N, P) = (M_i, G_i)_{i=1}^N \tag{1}$$

where for each frame $i$, $M_i \in \{0, 1\}^{H \times W}$ is the instance segmentation mask, and $G_i$ is a comprehensive **geometry set** defined as:

$$G_i = \{\mathbf{K}_i, \mathbf{X}_i, \mathbf{C}_i\} \tag{2}$$

Here, $\mathbf{K}_i \in SE(3) \subset \mathbb{R}^{4 \times 4}$ denotes the camera pose, $\mathbf{X}_i \in \mathbb{R}^{H \times W \times 3}$ represents the pixel-aligned 3D point map in the local camera coordinate system, and $\mathbf{C}_i \in [0, 1]^{H \times W}$ is the confidence map providing per-pixel reliability scores for the predicted geometry $\mathbf{X}_i$.

### 3.2. Architecture of G²TAM

As illustrated in Fig. 2, our G²TAM builds upon Pi3 (Wang et al., 2025c), and consists of three parts: 1) *Prompt Encoders* to accept various types of visual or text prompts, 2) *Cross-modal Spatial Encoder* to fuse the vision tokens and prompt tokens to construct a unified geometry-semantic representation, and 3) *Geometry & Mask Decoder* to predict the geometry of the scene and corresponding spatial-temporal consistent object masks for each frame, respectively.

**Prompt Encoding.** To facilitate flexible, multi-modal guidance, G²TAM projects diverse prompt types into a unified geometric-semantic latent space. Each prompt is encoded into the prompt tokens $\mathbf{T}_p \in \mathbb{R}^{N_p \times D}$, where $N_p$ varies according to the specific prompt modality:

1) *Visual Prompt Encoder:* Following SAM (Ma et al., 2024), we represent visual prompts through a combination of spatial coordinates and semantic identifiers. Specifically, a visual prompt $P_v$ at coordinate $(x, y)$ is transformed into a $D$-dimensional token $\mathbf{t}$ via:

$$\mathbf{t} = \text{PE}(x, y) + \mathbf{e}_{type} \tag{3}$$

where $\text{PE}(\cdot)$ denotes a positional encoding based on random Fourier features, and $\mathbf{e}_{type} \in \mathbb{R}^D$ is a learnable embedding distinct to each prompt type (e.g., positive/negative points or box corners). Consequently, a *point prompt* yields $N_p = 1$ token, while a *box prompt* is decomposed into its top-left and bottom-right corners, generating $N_p = 2$ tokens.

2) *Text Prompt Encoder:* Text prompts are tokenized with a pretrained tokenizer and processed through the CLIP (Radford et al., 2021) text encoder to obtain sentence-level embeddings, which are further transformed by a learnable projection layer to produce the final $N_p = 1$ text token.

**Cross-Modal Spatial Encoder.** Each image $\mathbf{I}_i$ is first embedded into vision tokens $V_i \in \mathbb{R}^{J \times D}$ via a DINOv2 (Oquab et al., 2023) backbone. To enable unified spatial reasoning, we define a composite input sequence $Z_i$ for each frame $i$:

$$Z_i = [V_i \; ; \; \mathbf{T}_{p,i} \; ; \; R] \tag{4}$$

where $R$ denotes learnable register tokens. For **visual prompts**, which are inherently frame-specific, we adopt a localized conditioning strategy: if a frame $i$ contains visual prompts, $\mathbf{T}_{p,i}$ is populated with the corresponding visual prompt tokens; otherwise, it is filled by the *learnable null-prompt embeddings* to maintain structural consistency. In contrast, for **text prompts**, which provide global semantic context, we adopt a consistent fusion strategy: the text tokens derived from the CLIP encoder are inserted as $\mathbf{T}_{p,i}$ into the sequences of every frame. The resulting sequences

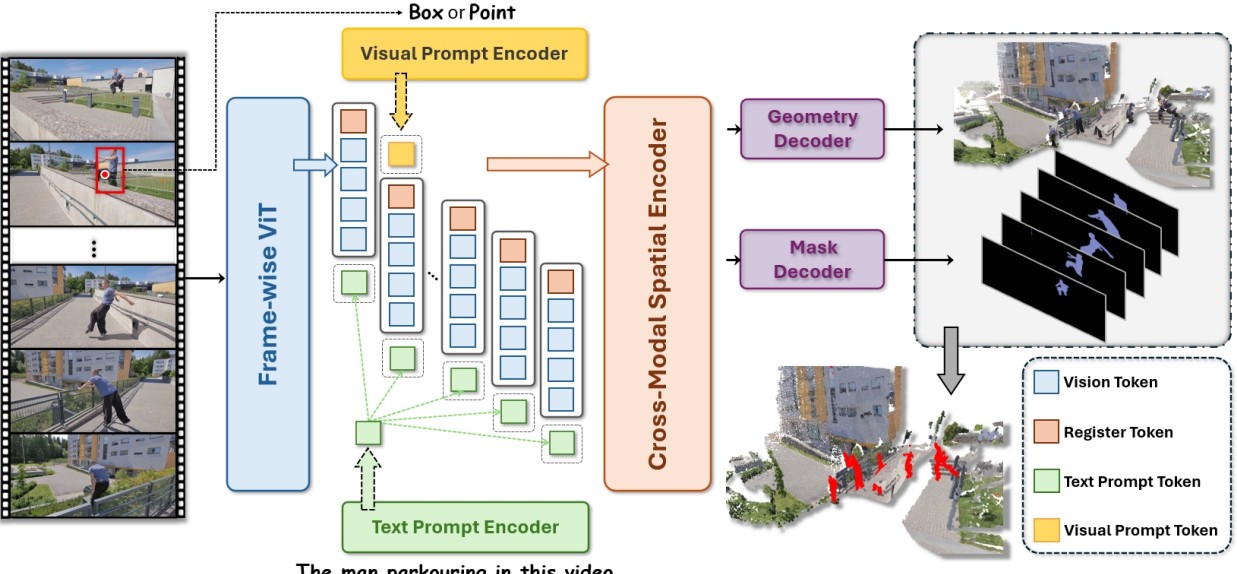

*Figure 2.* **Overview of G²TAM.** G²TAM is a unified framework for spatial reconstruction and promptable instance tracking with 2D prompts in 3D space from unposed RGB observations.

$\mathcal{Z} = (Z_i)_{i=1}^N$ are processed through alternating blocks of intra-view self-attention and global cross-view attention to obtain the fused geometric-semantic representation.

**Geometry & Mask Decoder.** The fused geometry-semantic representation from each frame is passed to two specialized decoders to produce the final predictions: 1) **Geometry Decoder**: Inherited from Pi3 (Wang et al., 2025c), this decoder predicts a comprehensive geometry set $G_i$ for each frame. 2) **Mask Decoder**: Adapted from SAM (Kirillov et al., 2023), this decoder predicts the instance mask $M_i$. Due to the early multi-modal fusion mechanism introduced by our cross-modal spatial encoder, we eliminate the prompt embeddings during mask decoding. Besides, we append an extra token alongside the mask and IoU tokens, and apply an additional MLP head to this token to predict the likelihood that the target object is present in the frame.

### 3.3. InsTrack Dataset and Benchmark

**Data Collection Pipeline.** To facilitate promptable instance tracking, we develop an automated annotation pipeline to generate multi-modal prompts and instance-consistent masks for the ScanNet++ dataset (Yeshwanth et al., 2023). The pipeline streamlines three stages: (1) *Image Subsampling*: We employ pose-based filtering (Zhang et al., 2025a) to remove ∼80% of redundant frames while maintaining maximal scene coverage. (2) *Mask Generation*: 2D instance masks are obtained by rasterizing 3D mesh annotations and establishing pixel-to-face mappings. (3) *Multi-modal Prompting*: We randomly sample up to 20

objects per scene to generate *visual prompts* ( points and boxes) and integrate existing human-annotated or reasoning-based *text prompts* from L3DD (Arnaud et al., 2025) and SURPRISE3D (Huang et al., 2025). We provide more data collection details in Appendix A.

**InsTrack Dataset.** Following the official ScanNet++ splits, the **InsTrack** dataset comprises 856 training and 50 validation scenes. Each sample provides RGB images, multi-modal prompts, depth maps, camera poses, and 3D-consistent instance masks. In total, the training set encompasses 99,666 text and 77,007 visual prompts (64,213 points, 12,794 boxes), while the validation set contains 3,784 text and 2,018 visual prompts (1,270 points, 748 boxes). We partition the benchmark into *InsTrack-Text* and *InsTrack-Visual* to individually assess the model's proficiency across different prompt modalities.

### 3.4. Model Training

To achieve promptable instance tracking capabilities while maintaining the original geometry reconstruction capabilities, our model is trained end-to-end by minimizing a composite loss function $\mathcal{L}$, which is a weighted sum of the segmentation loss $\mathcal{L}_{seg}$, and the geometry loss $\mathcal{L}_{geo}$, which consists of the point reconstruction loss, the confidence loss, and the camera pose loss:

$$\mathcal{L} = \lambda_{seg}\mathcal{L}_{seg} + \lambda_{geo}\mathcal{L}_{geo}. \tag{5}$$

$$\mathcal{L}_{geo} = \mathcal{L}_{points} + \lambda_{normal}\mathcal{L}_{normal} + \lambda_{conf}\mathcal{L}_{conf} + \lambda_{cam}\mathcal{L}_{cam}. \tag{6}$$

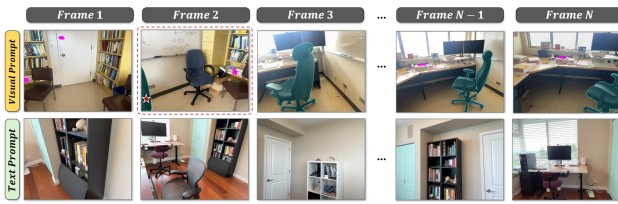

*Figure 3.* **Qualitative results on InsTrack validation set.** The top and bottom rows illustrate that our G²TAM effectively handles diverse prompt types and maintains spatially consistent instance tracking across frames.

To ensure generalization and robustness, we train the model on a large-scale aggregation of datasets, including segmentation datasets, reconstruction datasets, and joint segmentation-reconstruction datasets. Details of model training can be found in Appendix B.

# 4. Experiment

## 4.1. Promptable Instance Spatial Tracking

In this section, we introduce a new task: Promptable Instance Spatial Tracking (PIST), designed to evaluate a model's instance segmentation spatial consistency across views in the static scenario. Given an input prompt—such as a point, bounding box on any view, or a referring text expression—the goal is to produce spatially consistent masks for the corresponding instance across all views. For the PIST task, we evaluate performance using Spatial mIoU (S-mIoU) and Spatial Success Rate (S-SR). Given an object $o$ and its predicted masks $\{\hat{M}_n^o\}_{n=1}^N$ across $N$ views with ground-truth masks $\{M_n^o\}_{n=1}^N$, S-mIoU is defined as

$$\text{S-mIoU}(o) = \frac{1}{N} \sum_{t=1}^{N} \frac{|\hat{M}_n^o \cap M_n^o|}{|\hat{M}_n^o \cup M_n^o|} \tag{7}$$

S-SR measures whether the object is consistently tracked across all views. A frame is considered successfully tracked only if the frame IoU exceeds 0.5. Formally,

$$\text{S-SR}(o) = \mathbb{1}\!\!\!\!\mathbb{K} \left[ \forall n \in \{1, \dots, N\}, \frac{|\hat{M}_n^o \cap M_n^o|}{|\hat{M}_n^o \cup M_n^o|} > 0.5 \right] \tag{8}$$

We benchmark G²TAM against strong video object segmentation methods on the InsTrack validation set (Tab. 1). For InsTrack-*Visual*, we compare to Cutie (Cheng et al., 2024) and SAM2 (Ravi et al., 2024), while for InsTrack-*Text*, we include the recent referring video object segmentation models ReferFormer (Wu et al., 2022) and ReferDINO (Liang et al., 2025) to ensure a fair comparison. Across all settings, our G²TAM establishes a clear performance margin.

On the *Text* part, our method achieves 72.3 S-mIoU and 77.6 S-SR, substantially outperforming ReferFormer (37.6 / 43.7) and ReferDINO (41.7 / 48.2). On the *Visual* part, G²TAM further improves performance to 75.8 S-mIoU and 81.2 S-SR, surpassing SAM2 and Cutie-base by large margins. Aggregated over both subsets, our model attains 74.3 S-mIoU and 80.1 S-SR, marking a significant improvement over all baselines. These results highlight the benefit of using geometry-aware representations as implicit memory. Unlike conventional video segmentation models—which tend to lose track under substantial viewpoint change or extended temporal spans, often yielding S-mIoU scores below 50—G²TAM reliably maintains object identity across views and achieves strong spatial tracking success, with the qualitative results in Fig. 3.

## 4.2. 3D Visual Grounding

3D Visual Grounding aims to localize a target object in 3D scene given a natural-language referring expression. In this section, we study the performance of G²TAM on three standard visual grounding benchmarks: SR3D (Achlioptas et al., 2020), NR3D (Achlioptas et al., 2020), and ScanRefer (Chen et al., 2020). To unify the evaluation, we follow the previous method (Arnaud et al., 2025) to evaluate all the methods by reporting the top-1 accuracy without assuming ground-truth object bounding boxes under Acc@0.25 and Acc@0.5 metrics. G²TAM can directly reconstruct 3D scenes and predict instance-consistent masks across images. However, the reconstructed point clouds output by our model lack absolute scale information, making it difficult to perform a fair comparison in 3D space. To ensure accurate quantitative comparison, we therefore employ ground-truth depth and camera poses to project each predicted mask across views into 3D space to obtain 3D instance point clouds masks. In addition, to improve the quality of the final obtained 3D bounding boxes, we apply a post-processing stage that filters out noise points to determine the final 3D bounding box.

As shown in Tab. 2, our G²TAM significantly outperforms all previous zero-shot approaches on both NR3D and ScanRefer. In particular, it surpasses the prior state-of-the-art method, VLM-Grounder, by a large margin—achieving 45.7% overall Acc@0.5 compared to VLM-Grounder's 32.8% on ScanRefer. Notably, VLM-Grounder relies on a combination of a VLM with external detection and segmentation tools to obtain 2D object masks across views, which are then projected and fused into the 3D point cloud. This multi-stage process leads to a large performance gap (18.8) between Acc@0.25 and Acc@0.5. In contrast, our G²TAM inherently maintains geometric consistency, yielding far more stable view alignment and a smaller gap (10.5) between Acc@0.25 and Acc@0.5. Moreover, even without using explicit 3D geometry from point clouds as inputs,

| Model | Capability | | | Spatial Track (S-mIoU / S-SR) | | | Recon. Metric | |
|---|---|---|---|---|---|---|---|---|
| | Recon. | Understand | Track | *Text* | *Visual* | *Overall* | Abs. Rel↓ | $\tau$ ↑ |
| ReferFormer (Wu et al., 2022) | ✗ | ✓ | ✓ | 37.6 / 43.7 | - / - | 37.6 / 43.7 | - | - |
| ReferDINO (Liang et al., 2025) | ✗ | ✓ | ✓ | 41.7 / 48.2 | - / - | 41.7 / 48.2 | - | - |
| Cutie-base (Cheng et al., 2024) | ✗ | ✓ | ✓ | - / - | 42.7 / 51.9 | 42.7 / 51.9 | - | - |
| SAM2 (Ravi et al., 2024) | ✗ | ✓ | ✓ | - / - | 47.6 / 53.1 | 47.6 / 53.1 | - | - |
| VGGT (Wang et al., 2025a) | ✓ | ✗ | ✗ | - | - | - | 2.67 | 85.87 |
| Pi3 (Wang et al., 2025c) | ✓ | ✗ | ✗ | - | - | - | 2.54 | 86.72 |
| **$G^2$TAM (Ours)** | ✓ | ✓ | ✓ | **72.3 / 77.6** | **75.8 / 81.2** | **74.3 / 80.1** | **2.51** | **86.91** |

*Table 1.* **Quantitative results on InsTrack validation set.** We report the instance spatial tracking quality on the *Text* part, *Visual* part, and *Overall*, together with reconstruction accuracy metrics. **Bold** indicates the best results.

$G^2$TAM still outperforms the previous point-cloud-based methods, 3D-VisTA, which were trained and evaluated on the same dataset, demonstrating its strong capability to infer object relationships directly from 2D images.

To verify that the visual grounding results do not depend on ground-truth geometry at inference time, we further evaluate 3D projection using the depth and camera pose predicted by $G^2$TAM. As shown in Tab. 3, replacing ground-truth geometry with predicted geometry leads to only a small drop on ScanRefer and SR3D. This confirms that $G^2$TAM learns sufficiently accurate RGB-only geometry to support high-quality 3D instance grounding.

### 4.3. Promptable Video Object Segmentation

Since no existing video object segmentation benchmarks support both text and visual prompts, to evaluate our model's instance tracking performance in dynamic scenarios, we conduct experiments under two settings: (1) *Semi-supervised Video Object Segmentation* (Semi-supervised VOS), where the prompt is a ground-truth mask provided on the first frame, and (2) *Referring Video Object Segmentation* (RVOS), where the prompt is a natural-language referring expression. Unlike Promptable Instance Spatial Tracking, which primarily evaluates spatial consistency in static scenes, both Semi-supervised VOS and RVOS focus on dynamic scenarios where target objects undergo motion over time. These two settings require the model to preserve spatial accuracy and temporal coherence in segmentation predictions under complex motion and appearance variations, providing a comprehensive assessment of instance consistency in dynamic environments. For video object segmentation tasks, we report the performance using standard protocols J&F (Pont-Tuset et al., 2017).

We compare $G^2$TAM with prior approaches in Tab. 4 for Semi-supervised VOS and Tab. 5 for RVOS. As shown in Tab. 4, $G^2$TAM consistently outperforms SAM2 (Ravi et al., 2024). To ensure a rigorous and fair comparison, we eval-

uate both models at a consistent inference resolution of $512 \times 512$. Notably, our model achieves the most significant performance gain on the MOSE benchmark—specifically designed to assess tracking under heavy occlusion—where our score improves from 75.2 to **77**.**8**. This substantial margin underscores the efficacy of **geometric grounding**, which empowers $G^2$TAM to maintain robust object correspondence even when visual cues are severely degraded. On RVOS benchmarks (Tab. 5), $G^2$TAM achieves 72.2 $\mathcal{J\&F}$ on Ref-YTVOS and 71.7 on Ref-DAVIS17, significantly outperforming the previous SOTA ReferDINO (Liang et al., 2025) by +2.9 and +2.8, respectively. On the more challenging MeViS benchmark (Ding et al., 2023), which focuses on referring motion expressions, $G^2$TAM also improves over ReferDINO from 49.3/44.7/53.9 to 51.2/46.3/55.7 in $\mathcal{J\&F}/\mathcal{J}/\mathcal{F}$. Notably, $G^2$TAM performs these tasks without maintaining the explicit memory bank, demonstrating that the proposed global geometry representation inherently preserves instance correspondence across time, serving as a unified foundation for both static and dynamic visual understanding.

### 4.4. Inference Efficiency

We report matched inference statistics on a single A100 GPU using the same setting for SAM2 and $G^2$TAM ($512 \times 512$ input resolution with 8 frames). As shown in Tab. 6, $G^2$TAM introduces moderate overhead compared with SAM2, but this cost provides additional capabilities including RGB-only geometry prediction, text-conditioned spatial tracking, and 3D visual grounding. Profiling in Tab. 7 shows that the main cost comes from frame-wise attention and global cross-view attention, which are responsible for geometry-aware multi-view reasoning.

### 4.5. 3D Scene Reconstruction

For reconstruction evaluation, we follow Pi3 (Wang et al., 2025c) to adopt Absolute Relative Error (Abs. Rel) and the

| Method | w/o. LLM/VLM | w/o. PC | Zero-Shot | SR3D Acc@0.25 | SR3D Acc@0.5 | NR3D Acc@0.25 | NR3D Acc@0.5 | ScanRefer Acc@0.25 | ScanRefer Acc@0.5 |
|---|---|---|---|---|---|---|---|---|---|
| ScanRefer (Chen et al., 2020) | ✓ | ✗ | ✗ | - | - | - | - | 35.5 | 22.4 |
| LanguageRefer (Roh et al., 2022) | ✓ | ✗ | ✗ | 39.5 | - | 28.6 | - | - | - |
| InstanceRefer (Yuan et al., 2021) | ✓ | ✗ | ✗ | 31.5 | - | 29.9 | - | 40.2 | 32.9 |
| SAT-2D (Yuan et al., 2021) | ✓ | ✗ | ✗ | 35.4 | - | 31.7 | - | 44.5 | 30.1 |
| 3D-VisTA (Zhu et al., 2023) | ✓ | ✗ | ✗ | 56.5 | 51.5 | 47.7 | 42.2 | 51.0 | 46.2 |
| BUTD-DETR (Jain et al., 2022) | ✓ | ✗ | ✗ | 52.1 | - | 43.3 | - | 52.2 | 39.8 |
| OpenScene (Peng et al.) | ✓ | ✗ | ✓ | - | - | - | - | 13.2 | 6.5 |
| LLM-Grounder (Yang et al., 2024) | ✗ | ✗ | ✓ | - | - | - | - | 17.1 | 5.3 |
| VLM-Grounder () | ✗ | ✓ | ✓ | - | - | 48.0 | - | 51.6 | 32.8 |
| **G$^2$TAM (Ours)** | ✓ | ✓ | ✓ | **57.2** | **48.7** | **51.4** | **45.8** | **56.2** | **45.7** |

*Table 2.* **3D visual grounding results on SR3D, NR3D, and ScanRefer.** We evaluate top-1 accuracy on the validation set without any assumption of ground-truth proposals. G$^2$TAM outperforms previous LLM/VLM-assisted agent methods and point-cloud input-based methods in a zero-shot manner.

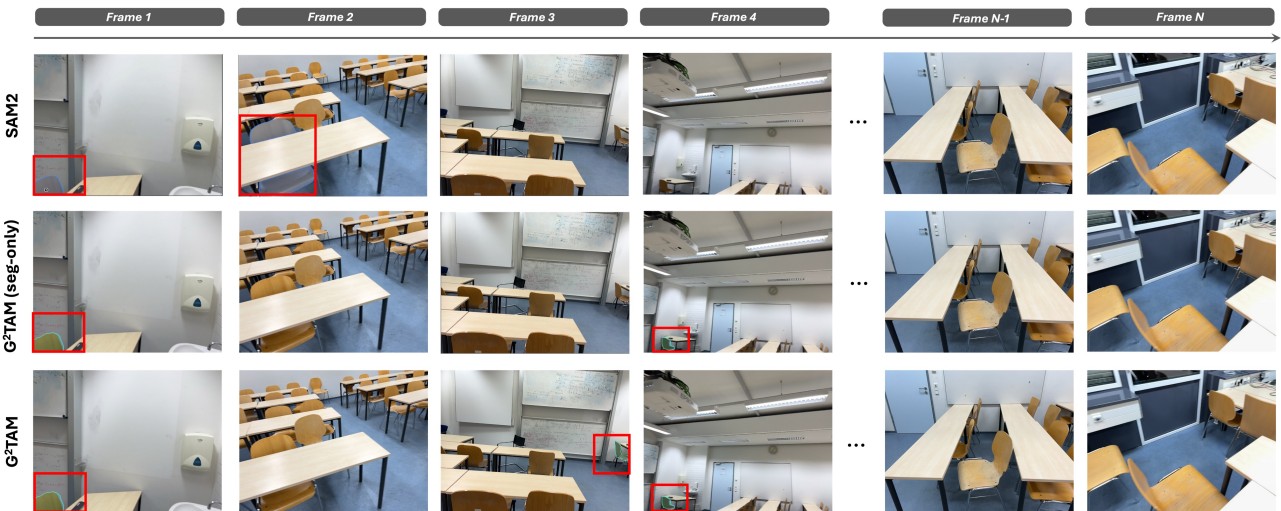

*Figure 4.* **Qualitative comparison of G$^2$TAM variants against SAM2.** By joint training on reconstruction and segmentation, G$^2$TAM achieves robust spatio-temporal consistency, maintaining precise tracking even in challenging sequences where baselines fail.

| 3D Projection Source | SR3D Acc@0.5 | ScanRefer Acc@0.5 |
|---|---|---|
| GT Depth + GT Pose | 48.7 | 45.7 |
| Pred. Depth + Pred. Pose | 47.3 | 44.9 |

*Table 3.* **3D visual grounding with predicted geometry.** G$^2$TAM remains close to the evaluation setting using ground-truth depth and pose, showing that it is not dependent on ground-truth geometry at inference time.

threshold accuracy metric ($\delta < 1.03$) for scene-level assessment. As shown in Tab. 1, our method attains lower Abs. Rel and higher $\delta$ accuracy than both Pi3 and VGGT on the InsTrack validation set. To further examine generalization, we evaluate scale-invariant monocular depth on the additional datasets using the standard Abs. Rel and $\delta < 1.25$ metrics. Tab. 8 demonstrates that our approach achieves

superior monocular depth accuracy compared to the Pi3.

## 4.6. Ablation Study

In this section, we systematically evaluate the core components of G$^2$TAM. We first establish a strong baseline, termed *SegPi3*, by simply extending the original Pi3 with a SAM-style mask decoder to support instance segmentation. Based on this, we analyze the impact of our architectural innovations and training strategies.

**Architecture Design and Text Integration.** We begin by justifying our architectural choice against the SegPi3 baseline. As shown in Tab. 9, SegPi3 treats text embeddings as sparse prompts injected directly into the decoder. However, this late-fusion approach results in a significant performance drop (from 72.3 to 61.8) on the InsTrack *Text*

| Method | $\mathcal{J}\&\mathcal{F}$ | | | | $\mathcal{G}$ |
|---|---|---|---|---|---|
| | MOSE val | DAVIS 2017 val | SA-V val | SA-V test | YTVOS 2019 val |
| STCN (Cheng et al., 2021) | 52.5 | 85.4 | 61.0 | 62.5 | 82.7 |
| SwinB-AOT (Yang et al., 2021) | 59.4 | 85.4 | 51.1 | 50.3 | 84.5 |
| SwinB-DeAOT (Yang & Yang, 2022) | 59.9 | 86.2 | 61.4 | 61.8 | 86.1 |
| RDE (Li et al., 2022) | 46.8 | 84.2 | 51.8 | 53.9 | 81.9 |
| XMem (Cheng & Schwing, 2022) | 59.6 | 86.0 | 60.1 | 62.3 | 85.6 |
| SimVOS-B (Wu et al., 2023) | - | 88.0 | 44.2 | 44.1 | 84.2 |
| JointFormer (Zhang et al., 2023) | - | 90.1 | - | - | 87.4 |
| ISVOS (Wang et al., 2023) | - | 88.2 | - | - | 86.3 |
| DEVA (Cheng et al., 2023) | 66.0 | 87.0 | 55.4 | 56.2 | 85.4 |
| Cutie-base (Cheng et al., 2024) | 69.9 | 87.9 | 60.7 | 62.7 | 87.0 |
| Cutie-base+ (Cheng et al., 2024) | 71.7 | 88.1 | 61.3 | 62.8 | 87.5 |
| SAM 2 (Ravi et al., 2024) | 75.2 | 89.4 | 75.8 | 76.7 | 87.8 |
| **G$^2$TAM (Ours)** | **77.8** | **89.9** | **76.8** | **77.6** | **89.1** |

*Table 4.* **Semi-supervised VOS results on various benchmarks.** G$^2$TAM achieves comparable performance with powerful SAM2 in accuracy ($\mathcal{J}\&\mathcal{F}$, $\mathcal{G}$) for video segmentation based on first-frame ground-truth mask prompts.

| Method | Ref-YouTube-VOS | | | Ref-DAVIS17 | | |
|---|---|---|---|---|---|---|
| | $\mathcal{J}\&\mathcal{F}$ | $\mathcal{J}$ | $\mathcal{F}$ | $\mathcal{J}\&\mathcal{F}$ | $\mathcal{J}$ | $\mathcal{F}$ |
| ReferFormer (Wu et al., 2022) | 62.9 | 61.3 | 64.6 | 61.1 | 58.1 | 64.1 |
| HTML (Han et al., 2023) | 63.4 | 61.5 | 65.2 | 62.1 | 59.2 | 65.1 |
| SgMg (Miao et al., 2023) | 65.7 | 63.9 | 67.4 | 63.3 | 60.6 | 66.0 |
| ReferDINO (Liang et al., 2025) | 69.3 | 67.0 | 71.5 | 68.9 | 65.1 | 72.9 |
| **G$^2$TAM (Ours)** | **72.2** | **69.1** | **73.1** | **71.7** | **68.2** | **75.1** |

*Table 5.* **RVOS results on various benchmarks.**

| Method | Peak Mem. (GB) | FPS | Time (s) |
|---|---|---|---|
| SAM2 (Ravi et al., 2024) | 3 | **30.2** | **0.26** |
| **G$^2$TAM (Ours)** | 4 | 21.6 | 0.37 |

*Table 6.* **Inference efficiency comparison with SAM2.** Both methods are evaluated on a single A100 GPU at $512 \times 512$ resolution with 8 input frames.

validation set. In contrast, our proposed cross-modal spatial encoder facilitates an early-fusion paradigm, demonstrating that integrating semantic guidance at the encoding stage is crucial for complex multi-modal reasoning. Building upon our early-fusion encoder, we further investigate the optimal strategy for token-level text injection. We experiment with inserting text tokens between vision and register tokens across varying numbers of frames ($1, 2, 5$, or all). Our results in Tab. 9 reveal that treating the text prompt as a global conditioning signal—by injecting it across all frames—achieves the best performance.

**Does Explicit Memory Help?** In this section, we investigate whether integrating a conventional explicit memory bank (as used in SAM2) could further bolster our model. We provide the full architectural details of this integration in Appendix C. Results in Tab. 10 show that explicit memory is not uniformly beneficial: it slightly improves SA-V

and MOSE, but degrades InsTrack *Visual* and DAVIS. This suggests that geometry-aligned implicit memory already captures most of the benefit, while combining it with explicit memory requires more careful design.

**How Geometry Benefits Segmentation?** We investigate whether the reconstruction objective truly bolsters spatio-temporal consistency in segmentation. First, replacing our Pi3-based encoder with a SAM2 encoder results in a significant performance drop (from 61.8 to 54.2) on InsTrack *Text* part, proving that geometry pretraining yields superior spatial alignment. Furthermore, we compare our joint-training paradigm against a segmentation-only baseline, G$^2$TAM (seg-only) in Tab. 11. Notably, joint training yields a substantial 4.0% improvement in segmentation accuracy. Besides, we provide a qualitative comparison between G$^2$TAM, G$^2$TAM (seg-only), and SAM2 to evaluate their tracking and segmentation capabilities in Fig. 4 . Given a visual prompt for a specific chair in Frame 1, SAM2 incorrectly segments a different chair in Frame 2 and fails to detect the target object in Frames 3 and 4. In contrast, G$^2$TAM (seg-only), which is trained exclusively on segmentation data, successfully identifies the target chair in Frame 4 but fails to accurately recognize it in Frame 3. However, only the full G$^2$TAM, leveraging the reconstruction objective as a geometric regularizer, maintains seamless tracking across all frames (e.g., Frames 3 and 4). This demonstrates that joint training effectively stabilizes the model's spatial memory, ensuring that promptable segmentation remains coherent and robust across diverse viewpoints.

## 5. Conclusion

In this paper, we presented G$^2$TAM, a unified framework for promptable instance tracking in 3D space, given only

| Component | Encoder | Frame-wise attention | Global cross-view attention | Mask head | Reconstruction head |
|---|---|---|---|---|---|
| **Time (s)** | 0.0720 | 0.0965 | 0.1395 | 0.0610 | 0.0521 |

*Table 7.* **Runtime breakdown of G²TAM.** The attention modules dominate inference time, indicating clear future directions for sparse or more efficient cross-view interaction.

| Method | Sintel | | KITTI | | NYU-v2 | |
|---|---|---|---|---|---|---|
| | Abs Rel↓ | $\delta < 1.25$ ↑ | Abs Rel↓ | $\delta < 1.25$ ↑ | Abs Rel↓ | $\delta < 1.25$ ↑ |
| DUSt3R (Wang et al., 2024) | 0.488 | 0.532 | 0.109 | 0.873 | 0.081 | 0.909 |
| MASt3R (Leroy et al., 2024) | 0.413 | 0.569 | 0.077 | 0.948 | 0.110 | 0.865 |
| MonST3R (Zhang et al., 2024) | 0.402 | 0.525 | 0.098 | 0.895 | 0.094 | 0.887 |
| Fast3R (Yang et al., 2025) | 0.544 | 0.509 | 0.120 | 0.861 | 0.093 | 0.898 |
| CUT3R (Wang et al., 2025b) | 0.418 | 0.520 | 0.097 | 0.914 | 0.081 | 0.914 |
| FLARE (Zhang et al., 2025b) | 0.606 | 0.402 | 0.312 | 0.513 | 0.089 | 0.898 |
| VGGT (Wang et al., 2025a) | 0.335 | 0.599 | 0.082 | 0.947 | 0.056 | 0.951 |
| Pi3 (Wang et al., 2025c) | 0.277 | 0.614 | 0.060 | 0.971 | 0.054 | 0.956 |
| **G²TAM (Ours)** | **0.275** | **0.616** | **0.059** | **0.973** | **0.052** | **0.959** |

*Table 8.* **Monocular Depth Estimation on Sintel (Bozic et al., 2021), KITTI (Geiger et al., 2013) and NYU-v2 (Silberman et al., 2012).**

| Method | Frame Num | InsTrack *Text* | Ref-YouTube-VOS |
|---|---|---|---|
| SegPi3 (baseline) | all | 61.8 | 63.2 |
| **G²TAM (Ours)** | 1 | 58.8 | 59.9 |
| | 2 | 63.2 | 61.7 |
| | 5 | 67.9 | 65.4 |
| | all | **72.3** | **70.2** |

*Table 9.* Ablation on architecture design and text integration strategies on InsTrack *Text* and Refer-YouTube-VOS benchmarks.

| Method | InsTrack *Visual* | SA-V val | MOSE | DAVIS |
|---|---|---|---|---|
| G²TAM | **75.8** | 73.9 | 77.8 | **89.9** |
| G²TAM + Memory Module | 74.9 | **74.2** | **78.2** | 89.5 |

*Table 10.* Ablation on explicit memory modules. All VOS columns report $\mathcal{J}\&\mathcal{F}$.

| Method | Training Data | InsTrack *Text* | InsTrack *Visual* |
|---|---|---|---|
| G²TAM (seg-only) | Seg | 68.2 | 72.7 |
| G²TAM | Seg + Recon | **72.3** | **75.8** |

*Table 11.* Ablation on joint reconstruction and segmentation training on InsTrack validation datasets.

significant societal benefits in automation and safety, we acknowledge the general ethical considerations associated with computer vision, including privacy concerns and the potential for misuse in surveillance. However, as this study focuses on fundamental algorithmic improvements using publicly available datasets, we do not foresee any immediate specific negative societal impacts.

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

unordered images or video as input. The key to our success is that we propose that spatially aligned geometric representations can serve as the implicit memory to achieve instance identity and localization across views and time in both static and dynamic scenes. To achieve this, we present a new promptable instance tracking dataset and benchmark, InsTrack, and the Promptable Instance Spatial Tracking (PIST) task. By joint training on segmentation and reconstruction datasets, our G²TAM could achieve robust instance tracking under large viewpoint variations and long-term sequences, facilitating a wide range of interactive and geometry-grounded applications.

## Acknowledgements

This work is partially supported by the National Natural Science Foundation of China (No. 62402406).

## Impact Statement

This work contributes to visual-spatial intelligence. By improving the spatial reasoning and temporal tracking capabilities of AI systems, our research has potential applications in areas such as autonomous driving, robotic manipulation, and augmented reality. While these advancements offer

mentation and tracking in egocentric videos. In *Proceedings of the Asian Conference on Computer Vision*, pp. 2562–2578, 2024.

Botach, A., Zheltonozhskii, E., and Baskin, C. End-to-end referring video object segmentation with multimodal transformers. In *Proceedings of the IEEE/CVF Conference on Computer Vision and Pattern Recognition*, pp. 4985–4995, 2022.

Bozic, A., Palafox, P., Thies, J., Dai, A., and Nießner, M. Transformerfusion: Monocular rgb scene reconstruction using transformers. *Advances in Neural Information Processing Systems*, 34:1403–1414, 2021.

Chen, D. Z., Chang, A. X., and Nießner, M. Scanrefer: 3d object localization in rgb-d scans using natural language. In *ECCV*, 2020.

Cheng, H. K. and Schwing, A. G. Xmem: Long-term video object segmentation with an atkinson-shiffrin memory model. In *European Conference on Computer Vision*, pp. 640–658. Springer, 2022.

Cheng, H. K., Tai, Y.-W., and Tang, C.-K. Rethinking space-time networks with improved memory coverage for efficient video object segmentation. *Advances in Neural Information Processing Systems*, 34:11781–11794, 2021.

Cheng, H. K., Oh, S. W., Price, B., Schwing, A., and Lee, J.-Y. Tracking anything with decoupled video segmentation. In *Proceedings of the IEEE/CVF International Conference on Computer Vision*, pp. 1316–1326, 2023.

Cheng, H. K., Oh, S. W., Price, B., Lee, J.-Y., and Schwing, A. Putting the object back into video object segmentation. In *Proceedings of the IEEE/CVF Conference on Computer Vision and Pattern Recognition*, pp. 3151–3161, 2024.

Dai, A., Chang, A. X., Savva, M., Halber, M., Funkhouser, T., and Nießner, M. Scannet: Richly-annotated 3d reconstructions of indoor scenes. In *Proceedings of the IEEE conference on computer vision and pattern recognition*, pp. 5828–5839, 2017.

Ding, H., Liu, C., He, S., Jiang, X., and Loy, C. C. Mevis: A large-scale benchmark for video segmentation with motion expressions. In *Proceedings of the IEEE/CVF international conference on computer vision*, pp. 2694–2703, 2023.

Furukawa, Y., Hernández, C., et al. Multi-view stereo: A tutorial. *Foundations and trends® in Computer Graphics and Vision*, 9(1-2):1–148, 2015.

Geiger, A., Lenz, P., Stiller, C., and Urtasun, R. Vision meets robotics: The kitti dataset. *The international journal of robotics research*, 32(11):1231–1237, 2013.

Han, M., Wang, Y., Li, Z., Yao, L., Chang, X., and Qiao, Y. Html: Hybrid temporal-scale multimodal learning framework for referring video object segmentation. In *Proceedings of the IEEE/CVF International Conference on Computer Vision*, pp. 13414–13423, 2023.

Hartley, R. and Zisserman, A. *Multiple view geometry in computer vision*. Cambridge university press, 2003.

Huang, J., Li, Z., Zhang, H., Chen, R., He, X., Guo, Y., Wang, W., Liu, T., and Gong, M. Surprise3d: A dataset for spatial understanding and reasoning in complex 3d scenes. *arXiv preprint arXiv:2507.07781*, 2025.

Jain, A., Gkanatsios, N., Mediratta, I., and Fragkiadaki, K. Bottom up top down detection transformers for language grounding in images and point clouds. In *ECCV*, 2022.

Kirillov, A., Mintun, E., Ravi, N., Mao, H., Rolland, C., Gustafson, L., Xiao, T., Whitehead, S., Berg, A. C., Lo, W.-Y., et al. Segment anything. In *Proceedings of the IEEE/CVF International Conference on Computer Vision*, pp. 4015–4026, 2023.

Langley, P. Crafting papers on machine learning. In Langley, P. (ed.), *Proceedings of the 17th International Conference on Machine Learning (ICML 2000)*, pp. 1207–1216, Stanford, CA, 2000. Morgan Kaufmann.

Leroy, V., Cabon, Y., and Revaud, J. Grounding image matching in 3d with mast3r. In *European Conference on Computer Vision*, pp. 71–91. Springer, 2024.

Li, F., Zhang, H., Sun, P., Zou, X., Liu, S., Yang, J., Li, C., Zhang, L., and Gao, J. Semantic-sam: Segment and recognize anything at any granularity. *arXiv preprint arXiv:2307.04767*, 2023.

Li, M., Hu, L., Xiong, Z., Zhang, B., Pan, P., and Liu, D. Recurrent dynamic embedding for video object segmentation. In *Proceedings of the IEEE/CVF Conference on Computer Vision and Pattern Recognition*, pp. 1332–1341, 2022.

Liang, T., Lin, K.-Y., Tan, C., Zhang, J., Zheng, W.-S., and Hu, J.-F. Referdino: Referring video object segmentation with visual grounding foundations. *arXiv preprint arXiv:2501.14607*, 2025.

Ma, J., He, Y., Li, F., Han, L., You, C., and Wang, B. Segment anything in medical images. *Nature Communications*, 15(1):654, 2024.

Miao, B., Bennamoun, M., Gao, Y., and Mian, A. Spectrum-guided multi-granularity referring video object segmentation. In *Proceedings of the IEEE/CVF International Conference on Computer Vision*, pp. 920–930, 2023.

Oquab, M., Darcet, T., Moutakanni, T., Vo, H., Szafraniec, M., Khalidov, V., Fernandez, P., Haziza, D., Massa, F., El-Nouby, A., et al. Dinov2: Learning robust visual features without supervision. *arXiv preprint arXiv:2304.07193*, 2023.

Pan, L., Baráth, D., Pollefeys, M., and Schönberger, J. L. Global structure-from-motion revisited. In *European Conference on Computer Vision*, pp. 58–77. Springer, 2024.

Peng, S., Genova, K., Jiang, C., Tagliasacchi, A., Pollefeys, M., Funkhouser, T., et al. Openscene: 3d scene understanding with open vocabularies. In *CVPR*.

Plizzari, C., Goel, S., Perrett, T., Chalk, J., Kanazawa, A., and Damen, D. Spatial cognition from egocentric video: Out of sight, not out of mind. In *2025 International Conference on 3D Vision (3DV)*, pp. 1211–1221. IEEE, 2025.

Pont-Tuset, J., Perazzi, F., Caelles, S., Arbeláez, P., Sorkine-Hornung, A., and Van Gool, L. The 2017 davis challenge on video object segmentation. *arXiv preprint arXiv:1704.00675*, 2017.

Radford, A., Kim, J. W., Hallacy, C., Ramesh, A., Goh, G., Agarwal, S., Sastry, G., Askell, A., Mishkin, P., Clark, J., et al. Learning transferable visual models from natural language supervision. In *International conference on machine learning*, pp. 8748–8763. PmLR, 2021.

Ranftl, R., Bochkovskiy, A., and Koltun, V. Vision transformers for dense prediction. In *Proceedings of the IEEE/CVF international conference on computer vision*, pp. 12179–12188, 2021.

Ravi, N., Gabeur, V., Hu, Y.-T., Hu, R., Ryali, C., Ma, T., Khedr, H., Rädle, R., Rolland, C., Gustafson, L., Mintun, E., Pan, J., Alwala, K. V., Carion, N., Wu, C.-Y., Girshick, R., Dollár, P., and Feichtenhofer, C. Sam 2: Segment anything in images and videos. *arXiv preprint arXiv:2408.00714*, 2024. URL https://arxiv.org/abs/2408.00714.

Roh, J., Desingh, K., Farhadi, A., and Fox, D. Languagerefer: Spatial-language model for 3d visual grounding. In *CoRL*, 2022.

Schönberger, J. L. and Frahm, J.-M. Structure-from-motion revisited. In *Conference on Computer Vision and Pattern Recognition (CVPR)*, 2016.

Schönberger, J. L., Zheng, E., Frahm, J.-M., and Pollefeys, M. Pixelwise view selection for unstructured multi-view stereo. In *European conference on computer vision*, pp. 501–518. Springer, 2016.

Seo, S., Lee, J.-Y., and Han, B. Urvos: Unified referring video object segmentation network with a large-scale benchmark. In *European conference on computer vision*, pp. 208–223. Springer, 2020.

Silberman, N., Hoiem, D., Kohli, P., and Fergus, R. Indoor segmentation and support inference from rgbd images. In *European conference on computer vision*, pp. 746–760. Springer, 2012.

Wang, J., Chen, D., Wu, Z., Luo, C., Tang, C., Dai, X., Zhao, Y., Xie, Y., Yuan, L., and Jiang, Y.-G. Look before you match: Instance understanding matters in video object segmentation. In *Proceedings of the IEEE/CVF conference on computer vision and pattern recognition*, pp. 2268–2278, 2023.

Wang, J., Chen, M., Karaev, N., Vedaldi, A., Rupprecht, C., and Novotny, D. Vggt: Visual geometry grounded transformer. In *Proceedings of the Computer Vision and Pattern Recognition Conference*, pp. 5294–5306, 2025a.

Wang, Q., Zhang, Y., Holynski, A., Efros, A. A., and Kanazawa, A. Continuous 3d perception model with persistent state. In *Proceedings of the Computer Vision and Pattern Recognition Conference*, pp. 10510–10522, 2025b.

Wang, S., Leroy, V., Cabon, Y., Chidlovskii, B., and Revaud, J. Dust3r: Geometric 3d vision made easy. In *Proceedings of the IEEE/CVF Conference on Computer Vision and Pattern Recognition*, pp. 20697–20709, 2024.

Wang, Y., Zhou, J., Zhu, H., Chang, W., Zhou, Y., Li, Z., Chen, J., Pang, J., Shen, C., and He, T. $\pi^3$: Permutation-equivariant visual geometry learning, 2025c. URL https://arxiv.org/abs/2507.13347.

Wu, J., Jiang, Y., Sun, P., Yuan, Z., and Luo, P. Language as queries for referring video object segmentation. In *Proceedings of the IEEE/CVF Conference on Computer Vision and Pattern Recognition*, pp. 4974–4984, 2022.

Wu, Q., Yang, T., Wu, W., and Chan, A. B. Scalable video object segmentation with simplified framework. In *Proceedings of the IEEE/CVF International Conference on Computer Vision*, pp. 13879–13889, 2023.

Yang, J., Chen, X., Qian, S., Madaan, N., Iyengar, M., Fouhey, D. F., and Chai, J. Llm-grounder: Open-vocabulary 3d visual grounding with large language model as an agent. In *ICRA*, 2024.

Yang, J., Sax, A., Liang, K. J., Henaff, M., Tang, H., Cao, A., Chai, J., Meier, F., and Feiszli, M. Fast3r: Towards 3d reconstruction of 1000+ images in one forward pass. In *Proceedings of the Computer Vision and Pattern Recognition Conference*, pp. 21924–21935, 2025.

Yang, Z. and Yang, Y. Decoupling features in hierarchical propagation for video object segmentation. *Advances in Neural Information Processing Systems*, 35:36324–36336, 2022.

Yang, Z., Wei, Y., and Yang, Y. Associating objects with transformers for video object segmentation. *Advances in Neural Information Processing Systems*, 34:2491–2502, 2021.

Yeshwanth, C., Liu, Y.-C., Nießner, M., and Dai, A. Scannet++: A high-fidelity dataset of 3d indoor scenes. In *Proceedings of the IEEE/CVF International Conference on Computer Vision*, pp. 12–22, 2023.

Yuan, H., Li, X., Zhang, T., Huang, Z., Xu, S., Ji, S., Tong, Y., Qi, L., Feng, J., and Yang, M.-H. Sa2va: Marrying sam2 with llava for dense grounded understanding of images and videos. *arXiv preprint arXiv:2501.04001*, 2025.

Yuan, Z., Yan, X., Liao, Y., Zhang, R., Wang, S., Li, Z., and Cui, S. Instancerefer: Cooperative holistic understanding for visual grounding on point clouds through instance multi-level contextual referring. In *ICCV*, 2021.

Zhang, J., Cui, Y., Wu, G., and Wang, L. Joint modeling of feature, correspondence, and a compressed memory for video object segmentation. *arXiv preprint arXiv:2308.13505*, 2023.

Zhang, J., Herrmann, C., Hur, J., Jampani, V., Darrell, T., Cole, F., Sun, D., and Yang, M.-H. Monst3r: A simple approach for estimating geometry in the presence of motion. *arXiv preprint arXiv:2410.03825*, 2024.

Zhang, J., Chen, Y., Zhou, Y., Xu, Y., Huang, Z., Mei, J., Chen, J., Yuan, Y.-J., Cai, X., Huang, G., et al. From flatland to space: Teaching vision-language models to perceive and reason in 3d. *arXiv preprint arXiv:2503.22976*, 2025a.

Zhang, S., Wang, J., Xu, Y., Xue, N., Rupprecht, C., Zhou, X., Shen, Y., and Wetzstein, G. Flare: Feed-forward geometry, appearance and camera estimation from uncalibrated sparse views. In *Proceedings of the Computer Vision and Pattern Recognition Conference*, pp. 21936–21947, 2025b.

Zhou, Y., Wang, Y., Zhou, J., Chang, W., Guo, H., Li, Z., Ma, K., Li, X., Wang, Y., Zhu, H., et al. Omniworld: A multi-domain and multi-modal dataset for 4d world modeling. *arXiv preprint arXiv:2509.12201*, 2025.

Zhu, Z., Ma, X., Chen, Y., Deng, Z., Huang, S., and Li, Q. 3d-vista: Pre-trained transformer for 3d vision and text alignment. In *ICCV*, 2023.

# A. InsTrack Dataset Generation

We create an automatic annotation pipeline to generate multi-modal prompts and instance-consistent masks for each frame from the ScanNet++ (Yeshwanth et al., 2023) dataset. The pipeline consists of three stages:

*(1) Image Subsampling.* Since the video frames provided by ScanNet++ contain a significant amount of redundant information, we adopt the image filtering method from SPAR (Zhang et al., 2025a), which leverages camera poses to reduce redundant images with high similarity. This approach effectively filters out approximately 80% of the redundant images, ensuring that only images with sufficiently distinct poses are retained, thereby achieving the best possible scene coverage with a limited number of frames.

*(2) 2D Instance Mask Generation.* Since ScanNet++ only officially provides 3D instance mesh annotations, we first rasterize the 3D meshes onto 2D images and save the 2D-3D mappings (pixel-to-face), which map image pixels to face indices. Then, we utilize this rasterization to obtain the instance annotations on the 2D images, thus constructing the corresponding instance ID map.

*(3) Multi-modal Prompting.* We curate a diverse set of visual and text prompts to guide instance tracking. For visual prompts, we select up to 20 objects per scene that appear in at least five subsampled frames. From the images containing a target object, we randomly choose one as the reference frame, where the instance mask provides the *mask prompt*. To enhance prompt diversity, we further sample five points within the mask as *point prompts*, and use the tight bounding box of the mask as the *box prompt*. For text prompts, we integrate a subset of annotations from L3DD (Arnaud et al., 2025) and SURPRISE3D (Huang et al., 2025), encompassing human-annotated referring expressions and complex language-guided queries that involve spatial, commonsense, and intentional reasoning.

# B. Training Details

To ensure robustness and broad applicability across indoor/outdoor and static/dynamic scenarios, we curate a large-scale three-part corpus to train $G^2$TAM. The corpus comprises (i) video object segmentation datasets: VOS (DAVIS, MOSE, YouTubeVOS), SA-V (Ravi et al., 2024), Ref-SAV (Yuan et al., 2025), and Ref-YTVOS (Seo et al., 2020), (ii) reconstruction datasets: ScanNet++ (Yeshwanth et al., 2023), ScanNet (Dai et al., 2017), ARKitScenes (Baruch et al., 2021), and OmniWorld (Zhou et al., 2025), and (iii) our joint segmentation–reconstruction dataset: InsTrack. We initialize the model with weights from Pi3 (Wang et al., 2025c) and fine-tune on the aggregated corpus. Training is conducted on 64 NVIDIA A100 GPUs using AdamW. We adopt differential learning rates for different model components: $6 \times 10^{-6}$ for the cross-modal spatial encoder and $1 \times 10^{-5}$ for the visual prompt encoder, geometry decoder, and mask decoder. We use the CLIP encoder (Radford et al., 2021) as the text prompt encoder and freeze it throughout training. To stabilize optimization, the data sampler ensures that each mini-batch contains only one data type. Our objective combines segmentation and geometry losses with weights $\lambda_{\text{seg}} = 2.0$ and $\lambda_{\text{geo}} = 1.0$, and the component weights within $\mathcal{L}_{\text{geo}}$ keeps the same with Pi3. The learning objective math formulation is provided in Appendix F.

# C. Architecture Details

**Frame-wise ViT Encoder.**   As outlined in the main paper, each input view is embedded into a sequence of patch-level vision tokens using DINOv2 (Oquab et al., 2023). To adapt the encoder for dense prediction tasks, we adopt a feature reassembly strategy inspired by DPT (Ranftl et al., 2021). Specifically, we extract features from transformer layers $l = \{5, 12\}$. Given that the ViT architecture is isotropic (i.e., features across layers share the same resolution), we employ convolutional layers to project the channel dimensions from 1024 to 32 and 64, respectively. Subsequently, these features are upsampled by factors of $4\times$ and $2\times$ to restore spatial resolution. Finally, these multi-scale features are integrated into the mask decoder via skip connections.

**Task Modes in Cross-Modal Spatial Encoder.**   We utilize Fig. 5 to illustrate the modes of our cross-modal spatial encoder when handling different types of prompts. *Visual Prompts (Left)*: As depicted in the left panel, when a visual prompt is provided on frame $t$, it is projected into a visual prompt embedding via the visual prompt encoder. This embedding is then inserted between the register tokens and the vision tokens extracted by DINOv2. For other frames without visual prompts, a learnable placeholder embedding is inserted at the corresponding position to maintain structural consistency. *Text Prompts (Right)*: In the case of text prompts, as shown in the right panel, the text prompt embeddings are inserted between the register tokens and vision tokens across all frames.

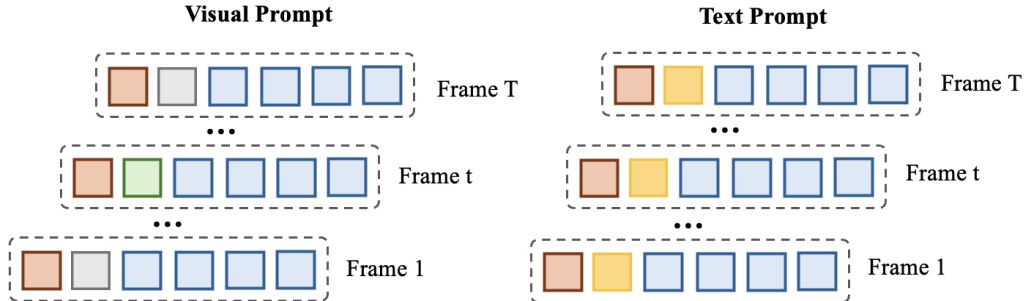

*Figure 5.* **Modes in Cross-Model Spatial Encoder.**

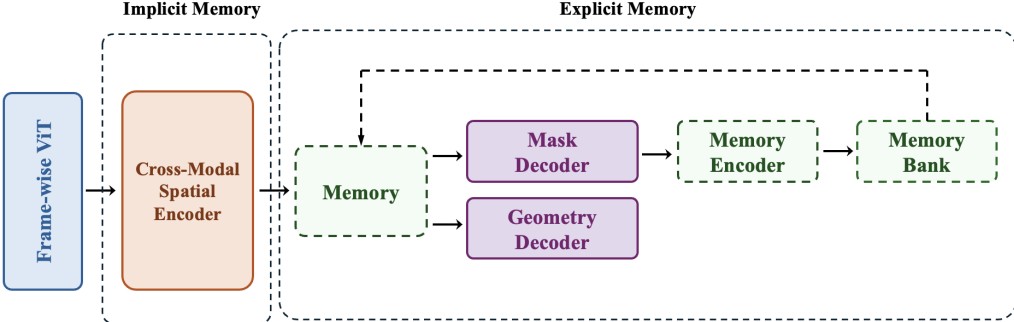

*Figure 6.* **Implicit and Explicit Memory Combination Architecture.**

**Architecture of Implicit and Explicit Combination.** In this section, we present the combination of implicit and explicit memory model architecture within our ablation study. Building upon the original G$^2$TAM architecture, we incorporate the memory mechanism from SAM2 (Ravi et al., 2024) by introducing additional memory encoder and memory bank modules (highlighted by the green dashed box in Fig. 6). Specifically, the memory encoder generates memory features by downsampling the output mask via a convolutional module. This representation is summed element-wise with the unconditioned frame embedding from the image encoder, followed by lightweight convolutional layers to fuse the information. The memory bank preserves temporal context by maintaining a First-In-First-Out (FIFO) queue of memories from the $N$ most recent frames. Consequently, within this explicit memory architecture, each frame prediction is conditioned not only on the current frame features but also on historical mask prediction results.

## D. More Visualization Results

We first present more visualization results of G$^2$TAM on the InsTrack validation set, which are primarily derived from L3DD (Arnaud et al., 2025). It is worth noting that the input images for these samples do not necessarily cover the entire scene; rather, they may focus on selected sub-regions within a larger environment. In this setting, our method demonstrates preliminary capabilities in spatial reasoning. For example, in the scene shown in the bottom row of Fig. 7, despite the presence of two keyboards, the model successfully identifies *"the keyboard closer to the window"* from the one that appeared in the first frame. Furthermore, the model exhibits remarkable cross-view consistency even across significant viewpoint changes.

We further extend our analysis to more challenging scenarios, as visualized in Fig. 8. In contrast to the previous samples, these images typically capture comprehensive views of the entire scene and involve significantly more complicated text prompts. Beyond simple pairwise spatial relationships, these prompts require **commonsense knowledge** (e.g., *object used for carrying items*), **logical reasoning**, and **spatial imagination** (e.g., *facing the bed with back to the wall*, *standing in front of the projection whiteboard*). Despite this complexity, G$^2$TAM accurately reasons through the instructions and consistently localizes the target objects across different frames. A notable example is shown in the bottom row: although the scene contains numerous chairs that satisfy the general description of *"a seating object"*, our model successfully disambiguates and identifies the specific chair that strictly satisfies all conditions. The segmented target objects are highlighted with red bounding boxes in the image.

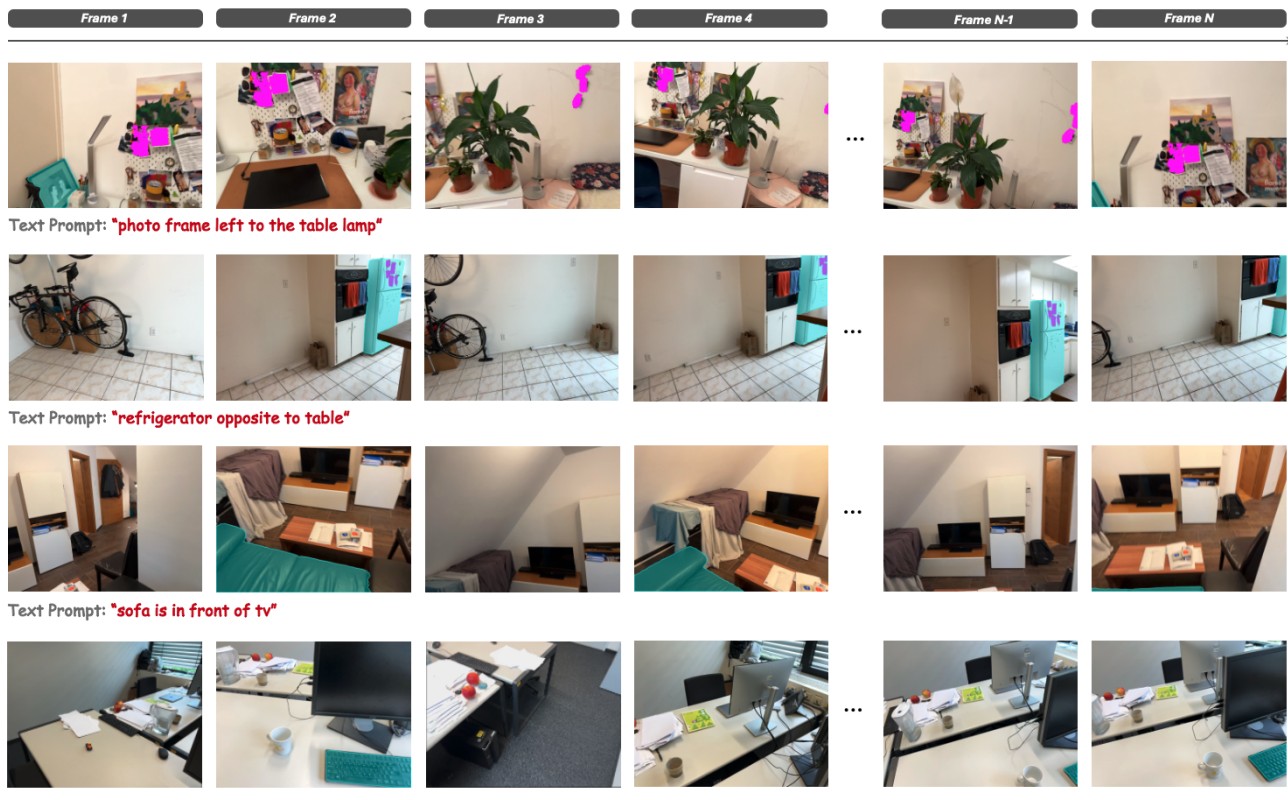

*Figure 7.* **Qualitative results on InsTrack validation set.** These samples typically focus on the sub-region of the large scene, and the text prompts typically consist of simple and short spatial relationships.

| Benchmark | Original Temporal Order | Shuffled Temporal Order |
|---|:---:|:---:|
| InsTrack *Visual* | **75.8** | 75.4 |
| InsTrack *Text* | **72.3** | 70.1 |

*Table 12.* Ablation on robustness to temporal permutation for InsTrack *Visual* and *Text*.

## E. More Spatial Memory Demonstration

**Explore and Revisit.** To further demonstrate the spatial memory capabilities of our model, we curated a set of long-term scenarios featuring an explore-and-revisit trajectory. In these sequences, the camera starts at a specific location, explores the environment for an extended duration, and subsequently returns to the initial scene. This setup simulates a realistic exploration mechanism. As shown in the Fig. 9, our model accurately comprehends the textual spatial descriptions in the initial frames (Frame 1 and Frame 2). Crucially, after a prolonged period of absence and exploration, when the camera returns to the starting point, our model successfully recalls the scene, accurately recognizing and segmenting the target objects despite the temporal gap.

**Robustness to Temporal Permutation.** Although our InsTrack dataset ensures significant viewpoint changes between frames, the sequences remain temporally consistent. To rigorously verify our model's spatial memory capabilities, we conducted an experiment on the InsTrack validation set where we randomly shuffled the input image sequences. This operation removes cues derived from visual temporal continuity, forcing the model to identify object consistency without relying on smooth temporal transitions.

Results in Tab. 12 indicate that performance remains largely consistent with visual prompts. However, we observe a slight performance degradation with text prompts; we hypothesize that this is because certain text descriptions may be implicitly

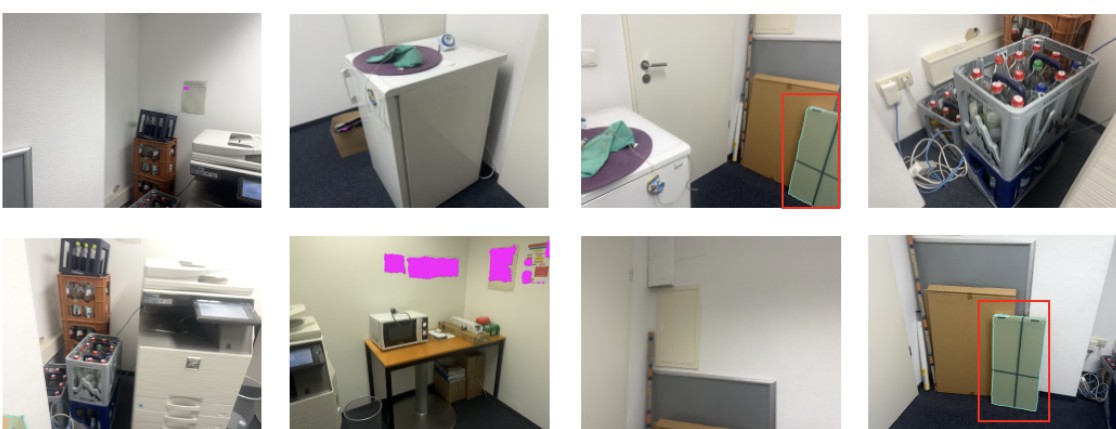

Prompt: **Back against the wall, facing the power distribution cabinet. Yellow object with black lines at the front left**

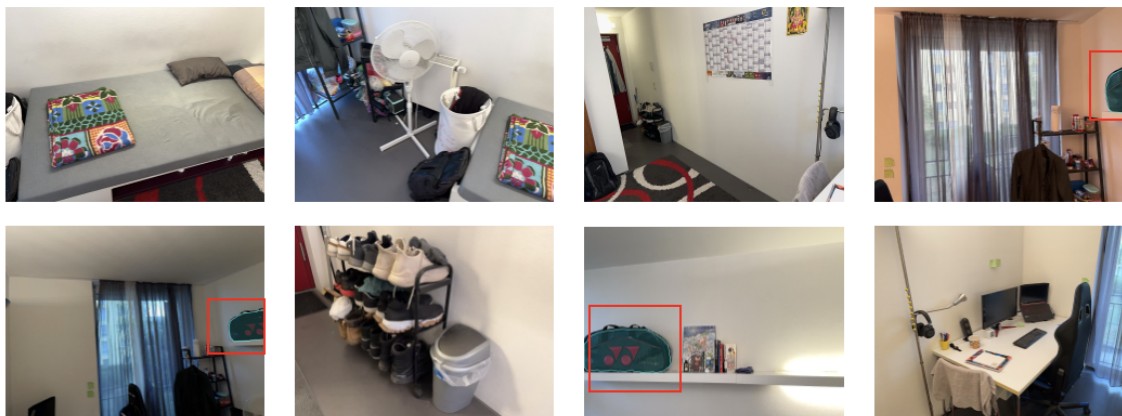

Prompt: **Facing the bed with back to the wall, upper left is a black object used for carrying items**

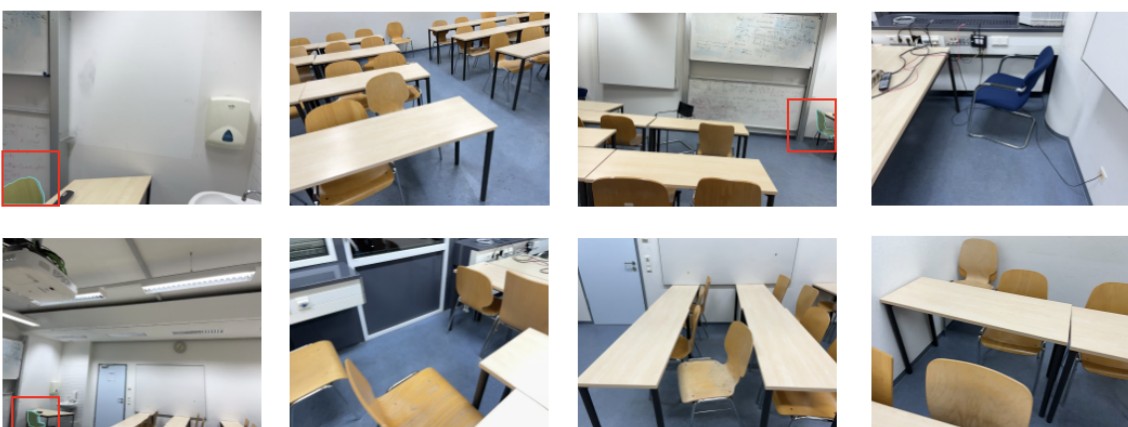

Prompt: **Standing in front of the projection whiteboard, facing away from it, with a seating object in the far left**

*Figure 8.* **Qualitative results on InsTrack validation set.** These samples cover the entire large scenes and the text prompts are much more complicated.

correlated with viewpoint continuity, making them sensitive to temporal disorder. Despite this, the model demonstrates overall robustness to temporal reordering, effectively exhibiting permutation equivariance (see Tab. 12). These findings confirm that our model relies primarily on spatial consistency across different viewpoints rather than temporal proximity in

Text Prompt: **"the chair close to the sofa"**

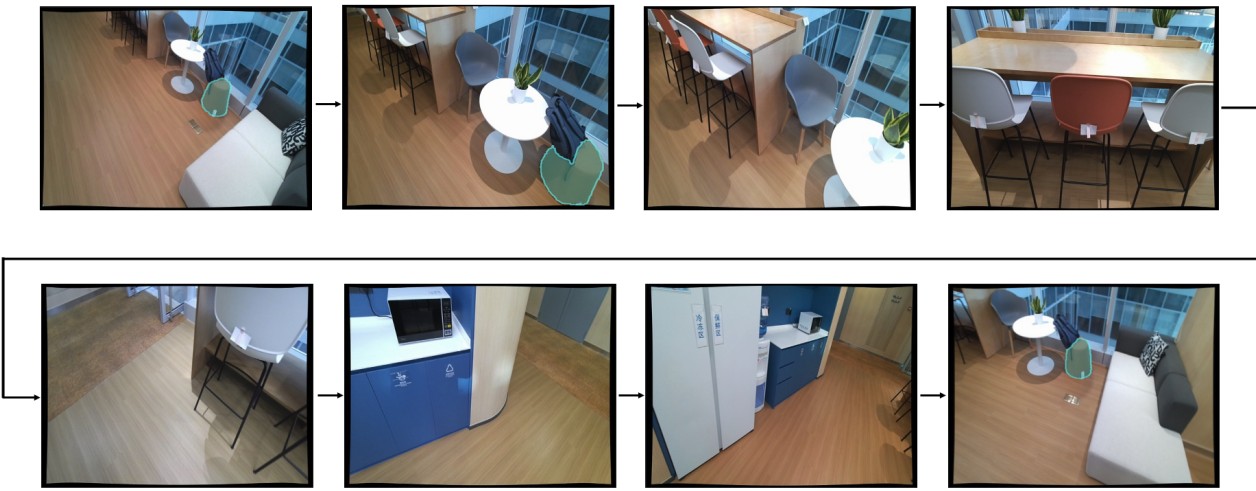

*Figure 9.* **Illustration of Explore-and-Revisit Trajectory.** Our G$^2$TAM maintains consistent object identity over long-term exploration, successfully re-recognizing targets during revisits.

static scenes, validating the effectiveness of our spatial memory mechanism.

## F. Learning Objective Details

To provide a rigorous mathematical foundation for the training process, we detail the formulation of each loss component. The total loss $\mathcal{L}$ is a weighted combination of the segmentation objective and the geometry objective:

$$\mathcal{L} = \lambda_{\text{seg}}\mathcal{L}_{\text{seg}} + \lambda_{\text{geo}}\mathcal{L}_{\text{geo}} \tag{9}$$

### F.1. Segmentation Loss

The segmentation loss $\mathcal{L}_{\text{seg}}$ ensures accurate 2D instance mask prediction for the target object. It consists of a weighted sum of Binary Cross-Entropy (BCE) and Dice losses:

$$\mathcal{L}_{\text{seg}} = w_{\text{bce}}\mathcal{L}_{\text{bce}} + w_{\text{dice}}\mathcal{L}_{\text{dice}} \tag{10}$$

where $w_{\text{bce}} = 2.0$ and $w_{\text{dice}} = 0.5$. Let $\hat{M}_i$ be the predicted mask logit in frame $i$ and $M_i$ be its ground-truth. The components are defined as:

$$\mathcal{L}_{\text{bce}} = -\frac{1}{N}\sum_{i=1}^{N}\frac{1}{HW}\sum_{j=1}^{H \times W}\left[M_{i,j}\log(\sigma_{i,j}) + (1 - M_{i,j})\log(1 - \sigma_{i,j})\right] \tag{11}$$

$$\mathcal{L}_{\text{dice}} = \frac{1}{N}\sum_{i=1}^{N}\left(1 - \frac{2\sum_{j}\sigma_{i,j}\cdot M_{i,j} + \epsilon}{\sum_{j}\sigma_{i,j} + \sum_{j}M_{i,j} + \epsilon}\right) \tag{12}$$

where $\sigma_{i,j} = \sigma(\hat{M}_{i,j})$ is the sigmoid activation and $\epsilon = 1.0$ is the smoothing constant.

### F.2. Geometry Loss

The geometry loss $\mathcal{L}_{\text{geo}}$ regularizes the 3D reconstruction and camera trajectory:

$$\mathcal{L}_{\text{geo}} = \mathcal{L}_{\text{points}} + \lambda_{\text{normal}}\mathcal{L}_{\text{normal}} + \lambda_{\text{conf}}\mathcal{L}_{\text{conf}} + \lambda_{\text{cam}}\mathcal{L}_{\text{cam}} \tag{13}$$

**Point Reconstruction and Scale Alignment.** To resolve scale ambiguity, we first solve for the optimal global scale factor $s^*$:

$$s^* = \arg\min_{s \in \mathbb{R}^+} \sum_{i,j} \frac{1}{z_{i,j}} \| s\hat{\mathbf{x}}_{i,j} - \mathbf{x}_{i,j} \|_1 \tag{14}$$

The reconstruction loss is then formulated as:

$$\mathcal{L}_{\text{points}} = \frac{1}{3NHW} \sum_{i,j} \frac{1}{z_{i,j}} \| s^*\hat{\mathbf{x}}_{i,j} - \mathbf{x}_{i,j} \|_1 \tag{15}$$

**Surface Normal and Confidence Loss.** The normal loss $\mathcal{L}_{\text{normal}}$ minimizes the angular distance between predicted and GT normals, while the confidence loss $\mathcal{L}_{\text{conf}}$ supervises the reliability map $\mathbf{C}_i$ via BCE loss against a precision-thresholded target.

**Camera Pose Loss.** The camera loss $\mathcal{L}_{\text{cam}}$ is defined over relative poses:

$$\mathcal{L}_{\text{cam}} = \frac{1}{N(N-1)} \sum_{i \neq j} \left( \mathcal{L}_{\text{rot}}(i,j) + \lambda_{trans} \mathcal{L}_{\text{trans}}(i,j) \right) \tag{16}$$

where $\mathcal{L}_{\text{rot}}$ is the geodesic distance on $SO(3)$ and $\mathcal{L}_{\text{trans}} = \mathcal{H}_\delta(s^*\hat{\mathbf{t}}_{i \leftarrow j} - \mathbf{t}_{i \leftarrow j})$ is the scale-rectified Huber loss.

