# OpenReview forum: "G$^2$TAM: Geometry Grounded Track Anything Model"
_ICML.cc/2026/Conference — ICML 2026 regular_

### Official Review · Reviewer_FVvv · 2026-03-01

**Soundness:** 2
**Presentation:** 3
**Significance:** 2
**Originality:** 3
**Overall Recommendation:** 4
**Confidence:** 3

**Summary:**

This paper proposes the G2TAM framework for promptable 3D instance tracking based on unordered RGB images or videos. The framework fuses appearance and geometric features into a shared spatial feature space, enabling stable instance tracking without explicit 3D inputs or memory banks. The model employs a cross-modal spatial encoder to process visual and textual prompts, enabling end-to-end spatial reconstruction and consistent mask prediction. Experiments validate G2TAM's performance across video object segmentation, 3D visual localization, and spatial tracking tasks. The paper also releases the InsTrack dataset for benchmarking these tasks.

**Compliance With Llm Reviewing Policy:**

Affirmed.

**Final Justification:**

My problem has been fully resolved, and I have adjusted my rating accordingly.

**Key Questions For Authors:**

I encourage the authors to carefully address the weaknesses outlined above. If additional empirical evidence, clarifications, or analyses can be provided to convincingly resolve these concerns, I would be open to revising my assessment and potentially raising the score accordingly.

**Limitations:**

yes

**Strengths And Weaknesses:**

Strengths：

1. The framework of geometric modeling as implicit memory has certain merit. This paper addresses the limitations of visual perception (VOS) methods that rely solely on appearance features and memory banks under conditions of large viewpoint changes and occlusion, and proposes corresponding solutions.

2. This paper fully elucidates the motivation behind the problem, provides a clear architectural description with illustrations, and explicitly defines the new task and evaluation metrics. Furthermore, this paper analyzes the training process in detail and provides architectural details in the appendix.

Weakness：

1. This model aggregates multi-frame information through a cross-view attention mechanism, but its streaming inference behavior, time window size limitations, and computational complexity for long sequences still require further investigation.

2. For 3D visual localization, the evaluation uses real depth and pose information to project masks into 3D space, which contradicts the description of "RGB-only" inference, making it difficult to evaluate the model's performance in predicting the geometry itself.

3. This paper does not compare with or discuss contemporary geometry-aware VOS methods, such as 3AM, nor does it discuss explicit 4D instance fields or 4D Gaussian feature fields (CIF, Feature4X), which illustrate the concept of "geometry as persistent memory" from different perspectives.

4. Running time, memory footprint, and inference scalability are not reported; given that training used 64 A100 processors, the actual deployment cost is unclear.

---

> ### Author Rebuttal · Authors · 2026-03-31
>
> ### W1. Streaming inference, window size limitations, and long-sequence scalability.
>
> We agree that streaming behavior and long-horizon deployment are important practical questions.
>
> However, the focus of this paper is **offline multi-frame aggregation**, not online causal inference. Our goal here is to evaluate whether the proposed geometry-aware aggregation mechanism is effective under a standard offline formulation. Extending such cross-view/global-attention models to true streaming inference typically requires dedicated modifications, such as **causal attention, KV-cached memory, or token/memory compression**, and is therefore a substantial research direction on its own rather than a minor extension of the present model.
>
> We will make this scope explicit in the revision and clearly state that streaming/long-sequence deployment is an important direction for future work.
>
> ### W2. 3D visual grounding uses GT depth/pose.
>
> Thank you for this clarification request. GT depth and camera pose are used **only for evaluation-time 3D projection**, following the standard benchmark protocol, not for inference. G²TAM itself predicts geometry from RGB inputs alone.
>
> We additionally evaluate 3D projection using **predicted depth + predicted pose**, and observe only a small drop:
>
> - **ScanRefer Acc@0.5:** 45.7 → 44.9
> - **SR3D Acc@0.5:** 48.7 → 47.3
>
> These results indicate that the model indeed learns high-quality geometry from RGB and is not dependent on GT geometry at test time.
>
> ### W3. Missing comparison/discussion with 3AM, CIF, and Feature4X.
>
> Thank you for highlighting these highly relevant geometry-aware works. We agree that they should be discussed more carefully, and we will add a dedicated paragraph in the related-work section to better position G²TAM.
>
> Among them, **3AM** is the closest work in spirit. It enhances **SAM2** by incorporating **3D-aware features from MUSt3R**, i.e., fusing **MUSt3R geometric features** with **SAM2 appearance features** through a feature-merging module. In this sense, 3AM is a highly relevant concurrent geometry-aware VOS direction. At the same time, it differs from G²TAM in two important ways. First, its evaluation is centered on challenging **wide-baseline datasets** such as **ScanNet++** and **Replica**, rather than the standard public VOS benchmarks used in our paper (e.g., DAVIS / MOSE / YouTubeVOS), and it does not report results for our text-prompted / referring-style setting. Therefore, an apples-to-apples benchmark comparison is not currently available from the paper itself. Second, from a deployment perspective, 3AM relies on features from **two encoders/backbones**—**MUSt3R** and **SAM2**—whereas G²TAM is designed as a unified feed-forward model for joint geometry-grounded tracking and reconstruction. To the best of our knowledge, we were also unable to obtain a publicly linked checkpoint or evaluation pipeline now, which further limited a controlled empirical comparison.
>
> By contrast, **CIF** and **Feature4X** are better viewed as **explicit 4D field-based dynamic scene representations** rather than benchmark-oriented feed-forward VOS methods. CIF builds a **consistent instance field** on top of **4D Gaussian Splatting** for panoptic segmentation and open-vocabulary 4D querying on dynamic scenes such as **HyperNeRF** and **Neu3D**. Feature4X similarly lifts pretrained 2D/video foundation-model capabilities into an explicit **4D Gaussian feature field**, targeting tasks such as novel-view segmentation, scene editing, and VQA over time. These works strongly support the broader intuition that persistent geometry/feature fields can serve as a form of long-term scene memory, but they differ substantially from G²TAM in formulation, supervision, and target tasks.
>
>
> ### W4. Runtime, memory footprint, and inference scalability.
>
> We agree that training cost and deployment cost should be clearly separated.
>
> The use of **64 A100 GPUs** refers **only to training**, i.e., large-scale optimization. It does **not** reflect inference-time deployment cost. For inference, under **512×512** resolution with **8 input frames**, G²TAM runs on **a single A100 GPU** at **21.6 FPS** with **4 GB** peak memory.
>
> | Component | Time (s) |
> |---|---:|
> | Encoder | 0.0720 |
> | Frame-wise attention | 0.0965 |
> | Global cross-view attention | **0.1395** |
> | Mask head | 0.0610 |
> | Reconstruction head | 0.0521 |
>
> Regarding scalability, runtime grows with the number of views, and the main bottleneck is the current **global cross-view attention** module. Our profiling shows that this overhead is concentrated in the attention components rather than in the heads, which suggests clear future optimization directions through sparse or more efficient cross-view attention. We will explicitly include this discussion in the revision.

---

> > ### Author Rebuttal · Reviewer_FVvv · 2026-04-02
> >
> > My problem has been fully resolved, and I have adjusted my rating accordingly.

---

> > > ### Author Response · Authors · 2026-04-02
> > >
> > > We sincerely thank the reviewer for the positive feedback and for raising the score, and we greatly appreciate the reviewer’s time, consideration, and encouraging assessment.

---

### Official Review · Reviewer_A5cg · 2026-03-06

**Soundness:** 3
**Presentation:** 3
**Significance:** 3
**Originality:** 3
**Overall Recommendation:** 4
**Confidence:** 3

**Summary:**

In this paper, given unordered RGB images or videos, the authors focus on promptable instance tracking in 3D, proposing a unified framework Geometry Grounded Tracking Anything Model (G$^2$TAM). The key idea of this paper is utilizing spatial aligned geometric representation as an implicit memory, which can bypass the need of explicit temporal memory bank or further post-hoc verification using explicit geometric representations. The experiments results show the effectiveness of the model in promptable instance spatial tracking, 3D Visual Grounding, and promptable video object segmentation.

**Compliance With Llm Reviewing Policy:**

Affirmed.

**Final Justification:**

I would like to thank the authors for the detailed and comprehensive rebuttal. I appreciate the effort put into addressing my concerns, particularly in running the new evaluations. I would like to keep my original rating.

**Key Questions For Authors:**

Please refer to the weakness part.

**Limitations:**

No. The model can only generate the 3D reconstructed point clouds without the metric information.

**Strengths And Weaknesses:**

Strengths:
1. The paper is well organized and easy to follow.
2. The main contribution in using unified geometric representation as an implicit memory, rather than using a temporal memory bank. This is a meaningful step for spatially-consistent tracking. The ablation show that G$^2$TAM’s implicit geometric memory is already sufficient for maintaining spatio-temporal consistency,
3. The dataset InsTrack, containing multi-modal prompts and instance-consistent masks built from the ScanNet++, is valuable to the research community.


Weaknesses:
1. This paper choose SAM2 as a benchmark for promptable video object segmentation; however, SAM2 is designed for highly efficient and real time segmentation. Therefore, the authors should provide the comparison in inference speed and memory footprint during inference.
2. The model is trained with the InsTrack dataset, which actually from ScanNet++, a dataset including static and indoor scenes. Therefore, how does the model also perform well in outdoor datasets, such as DAVIS and KITTI?

---

> ### Author Rebuttal · Authors · 2026-03-31
>
> ### W1. Inference speed and memory footprint compared with SAM2.
>
> Thank you for raising this important point. We now provide matched inference statistics on a single A100 GPU under the same setting (**512×512**, **8 frames**):
>
> | Method | Backbone | Peak Mem (GB) | FPS | Time |
> |---|---|---:|---:|---:|
> | SAM2 | Hiera-L | 3 | 30.2 | 0.26 s |
> | G²TAM | Pi3 + CLIP | 4 | 21.6 | 0.37 s |
>
> We also profile the runtime of each G²TAM component:
>
> | Component | Time (s) |
> |---|---:|
> | Encoder | 0.0720 |
> | Frame-wise attention | 0.0965 |
> | Global cross-view attention | **0.1395** |
> | Mask head | 0.0610 |
> | Reconstruction head | 0.0521 |
>
>
> The main additional cost comes from **frame-wise attention** and **global cross-view attention**, i.e., the modules responsible for geometry-aware multi-view reasoning, rather than from the output heads themselves.
>
> Importantly, this overhead buys capabilities that SAM2 does not provide: **RGB-only geometry prediction, text-conditioned spatial tracking, and 3D grounding with cross-view consistency**. Semi-supervised VOS alone understates the benefit of the joint formulation. We will clarify this trade-off more explicitly in the revision and discuss future acceleration directions such as more efficient attention, sparse cross-view interaction, and reduced-view computation.
>
> ### W2. Why does the model perform well on outdoor datasets if InsTrack comes from ScanNet++?
>
> InsTrack is only **one part** of our training corpus, not the sole training source.
>
> As stated in the appendix, our training data consists of three parts:
>
> 1. **Video object segmentation datasets:** DAVIS, MOSE, YouTubeVOS, SA-V, Ref-SAV, and Ref-YTVOS
> 2. **Reconstruction datasets:** ScanNet++, ScanNet, ARKitScenes, and OmniWorld
> 3. **Our joint segmentation–reconstruction dataset:** InsTrack
> Therefore, the model is trained on a mixture of **indoor and outdoor scenes**, as well as both segmentation and reconstruction data. Its performance on DAVIS/KITTI-style outdoor settings is thus not coming from ScanNet++ alone, but from this broader multi-domain training mixture. We will make this clearer in the paper.

---

> > ### Author Rebuttal · Reviewer_A5cg · 2026-04-03
> >
> > I would like to thank the authors for the detailed and comprehensive rebuttal. I appreciate the effort put into addressing my concerns, particularly in running the new evaluations. I would like to keep my original rating at this stage.

---

### Official Review · Reviewer_mUaT · 2026-03-13

**Soundness:** 3
**Presentation:** 4
**Significance:** 3
**Originality:** 3
**Overall Recommendation:** 5
**Confidence:** 3

**Summary:**

This paper presents G2TAM, a unified framework for promptable instance tracking that uses geometry-aligned representations as implicit memory, allowing more stable object identity and localization across viewpoints and time than appearance-based memory banks.  Its core technical idea is a cross-modal spatial encoder that fuses visual and text prompts into a shared geometric space for joint 3D reconstruction and instance-consistent mask prediction.  The paper also introduces the InsTrack dataset and benchmark to evaluate promptable spatial tracking under text and visual prompts.  Experiments show clear gains over prior methods, including the spatial tracking benchmark， suggesting strong robustness under viewpoint change and occlusion.

**Compliance With Llm Reviewing Policy:**

Affirmed.

**Final Justification:**

Thank you to the authors for the thoughtful rebuttal and the additional experiments. I believe the rebuttal has adequately addressed my main concerns, especially regarding reconstruction quality on Pi3 benchmarks, robustness to unordered inputs on public benchmarks, and the clarification of geometry as implicit memory. Overall, I find this work to be a solid and meaningful attempt to combine semantic information with geometric representations for promptable tracking, and I think this integration is both technically interesting and practically useful. Given the improved clarity and the strength of the empirical evidence after rebuttal, I remain supportive of acceptance.

**Key Questions For Authors:**

1. The paper describes its core idea as “geometry as implicit memory” to distinguish it from the explicit memory module used in SAM. However, it is not entirely clear why geometry should be characterized as memory in this context. The term “memory” may be more naturally suited to stream-based or sequential input settings, as in works such as CUT3R [1]. The authors should better justify this terminology and clarify in what sense the geometric representation functions as memory rather than simply as an internal scene representation.
2. Regarding the results in Fig. 9, why is text-prompt-based tracking noticeably more sensitive to temporal shuffling than visual-prompt-based tracking?

[1] Continuous 3D Perception Model with Persistent State

**Limitations:**

I believe the paper should discuss the efficiency trade-offs of different paradigms more explicitly, as integrating geometric features likely introduces substantial additional computational cost. In particular, it would be helpful to clarify whether the performance gains justify the added runtime and memory overhead compared with simpler appearance-based approaches.

**Strengths And Weaknesses:**

Strength:
1. The paper addresses an important weakness of appearance-based tracking and VOS systems: their limited robustness under large viewpoint changes and long-term occlusion. It is well motivated that geometry can serve as a more stable and reliable form of memory in these scenarios.
2. The paper thoroughly explores the advantages of the proposed network design, drawing inspiration from both SAM and geometric foundation models such as Pi3. These ideas are effectively integrated into G2TAM. The analysis is also valuable in discussing which design choices are truly necessary for this task and which components, such as explicit memory, may be redundant.
3. The introduction of InsTrack and the PIST task further strengthens the paper’s impact. The benchmark is explicitly designed for multi-prompt, cross-view, spatially consistent instance tracking and appears to be reasonably large in scale.
4. The proposed model demonstrates substantial gains on the new spatial tracking benchmark, achieving 74.3 S-mIoU overall. It also improves over prior methods on 3D visual grounding and outperforms strong baselines such as SAM2 and ReferDINO on VOS and RVOS benchmarks.

Weakness:
1. It would be helpful to also compare 3D point cloud reconstruction performance on the benchmark used in Pi3. In particular, I wonder whether incorporating semantic features may generally degrade 3D reconstruction quality on standard reconstruction benchmarks.
2. Equation 6 does not clearly explain each component of the composite loss. For example, the meaning of $L_{normal}$ should be described more clearly in the main paper.
3. A key missing evaluation is robustness to unordered inputs on standard public tracking benchmarks. The paper only studies temporal permutation on its own InsTrack benchmark, so it remains unclear whether G2TAM’s claimed advantage over methods such as SAM2 would still hold when all methods are evaluated under the same shuffled-order setting.

---

> ### Author Rebuttal · Authors · 2026-03-31
>
> ### W1. 3D reconstruction comparison on Pi3 benchmarks.
>
> Thank you for this suggestion. Following the Pi3 evaluation protocol, we additionally compare 3D reconstruction quality on the same standard benchmarks to test whether introducing semantic features harms geometric reconstruction.
>
> | | Camera Pose Estimation (angular) | | | Multi-View Reconstruction | | | | | |
> | :--- | :---: | :---: | :---: | :---: | :---: | :---: | :---: | :---: | :---: |
> | **Dataset** | **ScanNet** | | | **Scannet** | | | | | |
> | **Model** | **RRA@30 ↑** | **RTA@30 ↑** | **AUC@30 ↑** | **Acc. ↓** | | **Comp. ↓** | | **NC ↑** | |
> | pi3 | 96.74 | 92.46 | 73.18 | **0.0236** | **0.0137** | 0.0264 | 0.0153 | **0.7689** | **0.8787** |
> | G²TAM | **97.88** | **93.33** | **73.95** | 0.0246 | 0.0148 | **0.0259** | **0.0147** | 0.7645 | 0.8750 |
>
> The answer is no: G²TAM remains **comparable to Pi3**, and is **slightly better on several pose metrics**. The reconstruction metrics also remain at a comparable level, indicating that incorporating semantics for promptable tracking does **not** materially degrade the underlying geometric reconstruction quality.
>
>
> ### W2. Equation 6 clarification.
>
> We apologize for the insufficient explanation. In Eq. (6), **L_normal** denotes the angular loss between predicted and ground-truth surface normals. We will add this definition directly to the main paper and make the description of each loss term explicit. Full objective details are already provided in Appendix Section F, and we will improve the main-text clarity accordingly.
>
> ### W3. Robustness to unordered inputs on public benchmarks.
>
> This is an excellent suggestion. We now evaluate temporal permutation robustness not only on InsTrack, but also on **DAVIS** and **MOSE**:
>
> | Method | DAVIS Ordered | DAVIS Shuffled | MOSE Ordered | MOSE Shuffled |
> |---|---:|---:|---:|---:|
> | SAM2 | 89.4 | 81.2 | 75.2 | 67.2 |
> | G²TAM | 89.9 | 86.2 | 77.8 | 73.2 |
>
> Both methods degrade under temporal shuffling, but **G²TAM degrades substantially less**. This supports our claim that geometry-aligned representations provide stronger robustness to order disruption than appearance-based memory alone.
>
>
>
> ### Q1. Why call it “geometry as implicit memory”?
>
> We use the term **memory** in a **functional** sense, rather than a modality-specific sense.
>
> The learned geometry stores **persistent, scene-centered spatial information** that can be reused to re-identify and localize the target across viewpoint changes, occlusions, and time steps. In this role, it is analogous to the explicit appearance memory bank used by SAM-style trackers, but it stores geometry-aligned spatial state rather than appearance templates.
>
> We agree that this terminology should be explained more carefully. In the revision, we will clarify that “implicit memory” refers to the fact that past observations are retained and reused through a persistent geometric representation, rather than through an explicit memory-bank module.
>
> ### Q2. Why is text-prompt tracking more sensitive to temporal shuffling?
>
> Text prompts identify objects through **semantic descriptions**, which can remain spatially ambiguous when temporal order is disrupted (e.g., “the red chair near the window”). Without reliable temporal continuity, the model has less context to disambiguate among similar candidates.
>
> In contrast, visual prompts (point/box) provide an **explicit spatial anchor** in a specific frame. Once anchored, the geometry-aligned representation can propagate this localization across frames even under shuffled order. We will add this analysis to the paper.

---

> > ### Author Rebuttal · Reviewer_mUaT · 2026-04-04
> >
> > I Thank you to the authors for the thoughtful and detailed rebuttal. I appreciate the effort you put into clarifying several of my concerns. I will keep my original score.

---

### Official Review · Reviewer_fsJY · 2026-03-13

**Soundness:** 3
**Presentation:** 3
**Significance:** 3
**Originality:** 3
**Overall Recommendation:** 4
**Confidence:** 4

**Summary:**

This paper points out that existing memory-based VOS methods have difficulty tracking instances whose visibility changes under large viewpoint variations. To address this, the authors propose leveraging the geometric prior of a 3D Vision Foundation Model as implicit memory, replacing the reliance on explicit memory banks. The proposed framework jointly trains the geometry decoder and mask decoder for this task, and is fine-tuned to support both visual and textual prompt inputs. A new dataset, InsTrack, is constructed by curating existing data to facilitate training. The paper demonstrates that while slightly improving geometric reconstruction performance, VOS performance is substantially boosted, suggesting that implicitly incorporating geometric priors from 3D Vision Foundation Models can be beneficial for instance tracking.

**Compliance With Llm Reviewing Policy:**

Affirmed.

**Final Justification:**

The paper presents a technically sound and well-motivated approach to integrating 3D vision foundation models into video object segmentation. The idea of leveraging geometry-aligned representations as implicit memory is interesting, and the empirical results consistently demonstrate improvements across multiple benchmarks. The paper is generally well-structured, and the experimental section is thorough, including ablations and evaluations on both Semi-VOS and RVOS settings.

In my initial review, I had concerns regarding the clarity of the architectural contribution, the completeness of comparisons (especially with recent methods such as SAM3 and MeViS), and the justification of certain design choices. The rebuttal addresses these concerns to a large extent by providing additional experimental results, clarifying the role of the proposed cross-modal spatial encoder, and expanding comparisons on relevant benchmarks. The added efficiency analysis and extended ablations also improve the overall clarity of the method.

That said, I still find that the overall architectural novelty is somewhat limited, as the approach can be interpreted as an extension of existing 3D foundation models with additional components for segmentation and prompting. However, the integration is well-executed and practically meaningful, and the empirical gains are consistent and convincing.

Overall, considering both the paper and the rebuttal, I believe this work makes a solid contribution to the intersection of 3D vision and video object segmentation. The rebuttal has addressed most of my major concerns and improved my confidence in the paper, and I have adjusted my recommendation accordingly.

**Key Questions For Authors:**

1. Since G²TAM outputs 3D geometric representations, we expected segmentation to be performed in 3D space. However, in the 3D Visual Grounding experiments (Table 2), the evaluation relies on GT depth and camera poses. This makes the model appear to be essentially solving a 2D Video Object Segmentation problem. So, request clarification on what problem the authors ultimately aim to solve by producing 3D geometric output.

2. In Table 7, explicit memory slightly improves performance on SA-V (73.9→74.2), the only external benchmark used. Could the authors provide results on additional established benchmarks (e.g., MOSE, DAVIS) to more convincingly support the claim that explicit memory is unnecessary?

3. G²TAM uses Pi3, a substantially large model, together with CLIP. As shown in Table 8, including the geometry decoder in training (Seg+Recon) yields only approximately 4% improvement over Seg-only, and as shown in Table 3, the Semi-supervised VOS performance improves by only about 2% over SAM2. Is this an appropriate trade-off when compared to SAM2 in terms of computational speed and memory? So, request the authors to provide inference time, FLOPs, and peak GPU memory comparisons against SAM2.

4. Recent benchmarks such as MeViS[1] and recent models such as SAM3[2] are absent from the comparisons. Given the capacity of the proposed model, a direct comparison with SAM3 in particular seems necessary.

[1] Ding, Henghui, et al. "Mevis: A large-scale benchmark for video segmentation with motion expressions." Proceedings of the IEEE/CVF international conference on computer vision. 2023.

[2] Carion, Nicolas, et al. "Sam 3: Segment anything with concepts." arXiv preprint arXiv:2511.16719 (2025).

**Limitations:**

Yes

**Strengths And Weaknesses:**

Strengths

1. This is the first work to integrate a 3D vision foundation model (Pi3) as the backbone for Video Object Segmentation(VOS), achieving state-of-the-art performance on both Semi-VOS and RVOS benchmarks.
2. The authors systematically curate existing resources (ScanNet++, L3DD, SURPRISE3D) to construct the InsTrack dataset, along with a new task Promptable Instance Spatial Tracking(PIST) and evaluation metric (S-mIoU,S-SR).
3. The ablation studies are well-structured, progressively validating the effect of early fusion (Table 6) and the benefit of joint learning (Table 8).
4. The joint learning of geometry and segmentation decoder is shown to benefit both 3D Geometric prediction and VOS performance

Weaknesses
1. The reference for VLM-Grounder is missing in Table 2, and the dataset name position appears shifted, indicating editorial errors.
2. Tables 6-9. do not explicitly indicate which metrics are being reported in their column headers. Labeling the metrics like Table 1-5 would improve readability.
3. The approach appears to be adding a mask output on top of a 3D vision foundation model's output, combined with existing prompt engineering techniques from SAM. There appears to be a lack of contribution in terms of architecture
4. The comparisons omit recent relevant works such as MeViS and SAM3. Given the scale and ambition of the proposed framework, the absence of these comparisons limits the assessment of where G²TAM stands relative to the current state of the art
5. The comparison with existing memory-based methods is insufficient. The explicit memory experiment in Table 7 is limited to only two benchmarks, one of which (InsTrack Visual) is the authors' own proposed benchmark.

---

> ### Author Rebuttal · Authors · 2026-03-31
>
> ### W1. Missing reference and editorial errors.
>
> Thank you for pointing this out. We will fix the missing VLM-Grounder reference in Table 2 and correct the shifted dataset name in our final version.
>
> ### W2. Tables 6–9 missing metric labels.
>
> We agree that this hurts readability. We will add explicit metric labels to all column headers in Tables 6–9, including **S-mIoU** for InsTrack Text/Visual and **J&F** for Ref-YouTube-VOS and SA-V.
>
> ### W3. Lack of architectural contribution.
>
> We respectfully disagree that our method is merely a mask head added on top of a 3D foundation model, which actually refers to our baseline, SegPi3. Our proposed cross-modal spatial encoder injects **visual and textual prompts directly into the shared geometric feature space** through learned spatial attention. This is different from SAM-style prompt engineering, where prompts are used to guide 2D segmentation but are not integrated into a geometry-aligned representation. Empirically, this is not a marginal change: compared with the SegPi3 baseline, adding our spatial encoder improves performance from **61.8 → 72.3** on **InsTrack-Text** and from **63.2 → 70.2** on **Ref-YouTube-VOS**.
>
> ### W4 / Q4. Comparison with SAM3 and MeViS.
>
> The text interface of SAM3 is designed primarily for short noun-phrase concept prompts, whereas our text-prompted instance tracking requires reasoning over **complex referring expressions** for the referred instance. Therefore, a direct comparison in our text-conditioned tracking setting on our InsTrack and RVOS is not apples-to-apples. We compare against SAM3 on the **semi-supervised VOS** setting under the same input resolution (**512×512**). G²TAM is competitive with or better than SAM3 on most benchmarks, while trailing slightly on DAVIS.
>
> | Method | MOSE J&F | DAVIS J&F | YTVOS 2019 val J |
> |---|---:|---:|---:|
> | SAM2 | 75.2 | *89.4* | 87.8 |
> | SAM3 | *76.4* | **90.4** | *88.7* |
> | G²TAM | **77.8** | 89.9 | **89.1** |
>
> MeViS is a large-scale benchmark for **referring motion expression video segmentation**. Since SAM2/SAM3 cannot be directly applied to this setting, we instead compare with the current strong baseline **ReferDINO** on this benchmark. G²TAM significantly outperforms ReferDINO across the reported metrics.
>
> | Method | Metric 1 | Metric 2 | Metric 3 |
> |---|---:|---:|---:|
> | ReferDINO | 49.3 | 44.7 | 53.9 |
> | G²TAM | **51.2** | **46.3** | **55.7** |
>
> ### W5 / Q2. Explicit memory comparison on more benchmarks.
>
> We have expanded the explicit-memory ablation to additional established benchmarks.
>
> | Memory Type | MOSE J&F | DAVIS J&F |
> |---|---:|---:|
> | G²TAM | 77.8 | **89.9** |
> | G²TAM + Memory Module | **78.2** | *89.5* |
>
> These results suggest that explicit memory may not be consistently beneficial once a strong geometry-aligned representation is available. We therefore revise our claim to a more precise one: **geometry-aligned implicit memory already captures most of the benefit, while combining it with explicit memory requires more careful design and is not uniformly helpful.**
>
> ### Q1. 3D visual grounding relies on GT depth and pose.
>
> Thank you for this important clarification request. Our model does **not** require GT depth or GT camera poses at inference time when performing 3D visual grounding. In Table 2, GT depth and poses are used **only during evaluation**, following the standard protocol of prior 3D visual grounding benchmarks. To further clarify this point, we additionally evaluate 3D projection using **predicted depth + predicted pose** from our model. The performance drop is small:
>
> - **ScanRefer Acc@0.5:** 45.7 → 44.9
> - **SR3D Acc@0.5:** 48.7 → 47.3
>
> This shows that G²TAM is not merely solving a 2D problem; it learns sufficiently accurate geometry from RGB to support high-quality 3D instance grounding.
>
> ### Q3. Efficiency trade-off vs. SAM2.
>
> We agree that efficiency should be discussed explicitly. We now report matched inference statistics on **a single A100 GPU**, with the **same setting** for both models (**512×512**, **8 frames**):
>
>
> | Method | Backbone | Peak Mem (GB) | FPS | Time |
> |---|---|---:|---:|---:|
> | SAM2 | Hiera-L | 3 | 30.2 | 0.26 s |
> | G²TAM | Pi3 + CLIP | 4 | 21.6 | 0.37 s |
>
> We also profile the runtime of each G²TAM component:
>
> | Component | Time (s) |
> |---|---:|
> | Encoder | 0.0720 |
> | Frame-wise attention | 0.0965 |
> | Global cross-view attention | **0.1395** |
> | Mask head | 0.0610 |
> | Reconstruction head | 0.0521 |
>
> The main additional cost comes from **frame-wise attention** and **global cross-view attention**. Importantly, this overhead buys capabilities that SAM2 does not provide: **RGB-only geometry prediction, text-conditioned spatial tracking, and 3D visual grounding**. Evaluating the model solely on semi-supervised VOS understates the full advantages of our joint formulation. We will explicitly clarify this capability-efficiency trade-off in the final version and discuss future directions for acceleration.

---

> > ### Author Rebuttal · Reviewer_fsJY · 2026-04-02
> >
> > The rebuttal has addressed my concerns satisfactorily. I appreciate the authors’ clear and detailed responses. Based on these clarifications, I am convinced by the improvements and will adjust my score accordingly.

---

> > > ### Author Response · Authors · 2026-04-02
> > >
> > > We sincerely thank the reviewer for the positive feedback. We are glad that our clarifications addressed the concerns satisfactorily, and we greatly appreciate the reviewer’s time and consideration.

---

### Decision · Program_Chairs · 2026-04-30

**Decision:**

Accept (regular)

**Comment:**

This paper received positive reviews with scores of 4, 5, 4, and 4. All reviewers acknowledged that the rebuttal effectively addressed their concerns, including design choices, benchmarking, robustness to unordered inputs, and ablations. The provided efficiency analysis and extended ablations improved the overall clarity of the method. While there remained concerns about the overall architectural novelty, the reviewers considered the integration of existing 3D foundation models with additional components for segmentation and prompting to be both technically interesting and practically useful.

The area chair supports the reviewers' positive evaluations and recommends that the paper be accepted.